



# Modelling the influence of biotic plant stress on atmospheric aerosol
# particle processes throughout a growing season
Ditte Taipale[1,2,3,4], Veli-Matti Kerminen[1], Mikael Ehn[1], Markku Kulmala[1], Ülo Niinemets[2]
[1]Institute for Atmospheric and Earth System Research / Physics, Faculty of Science, University of Helsinki, P.O. Box 64,
00014 Helsinki, Finland
[2]Institute of Agricultural and Environmental Sciences, Estonian University of Life Sciences, Kreutzwaldi 1, Tartu 51006,
Estonia
[3]Institute for Atmospheric and Earth System Research / Forest Sciences, Faculty of Agriculture and Forestry, University of
Helsinki, PO Box 27, 00014 Helsinki, Finland
[4]Hyytiälä Forestry Field Station, Hyytiäläntie 124, 35500 Korkeakoski, Finland
*Correspondence to*: Ditte Taipale (ditte.taipale@helsinki.fi)
**Abstract.** Most trees emit volatile organic compounds (VOCs) continuously throughout their life, but the rate of emission,
and spectrum of emitted VOCs, become substantially altered when the trees experience stress. Still, models to predict the
emissions of VOCs do not account for perturbations caused by biotic plant stress. Considering that such stresses have generally
been forecast to increase in both frequency and severity in future climate, the neglect of plant stress-induced emissions in
models might be one of the key obstacles for realistic climate change predictions, since changes in VOC concentrations are
known to greatly influence atmospheric aerosol processes. Thus, we constructed a model to study the impact of biotic plant
stresses on new particle formation and growth throughout a full growing season. We simulated the influence on aerosol
processes caused by herbivory by European gypsy moth (*Lymantria dispar*) and autumnal moth (*Epirrita autumnata*) feeding
on pedunculate oak (*Quercus robur*) and mountain birch (*Betula pubescens* var. *pumila*), respectively, and also fungal
infections of pedunculate oak and balsam poplar (*Populus balsamifera* var. *suaveolens*) by oak powdery mildew (*Erysiphe
alphitoides*) and poplar rust (*Melampsora larici-populina*), respectively. Our modelling results indicate that all the investigated
plant stresses are capable of substantially perturbing both the number and size of aerosol particles in atmospherically relevant
conditions, with increases in the amount of newly formed particles by up to about one order of magnitude and additional daily
growth of up to almost 50 nm. We also showed that it can be more important to account for biotic plant stresses in models than
significant variations in e.g. leaf area index, and temperature and light conditions, which are currently the main parameters
controlling predictions of VOC emissions. Our study, thus, demonstrates that biotic plant stress can be highly atmospherically
relevant and it supports biotic plant stress emissions to be integrated into numerical models for prediction of atmospheric
chemistry and physics, including climate change projection models.
**1 Introduction**
Formation and subsequent growth of atmospheric aerosol particles is globally a major source of cloud condensation nuclei
(CCN) (Spracklen et al., 2008; Merikanto et al., 2009; Dunne et al., 2016). CCN impact various cloud processes, such as cloud
formation, albedo and lifetime (Twomey, 1977; Albrecht, 1989; Makkonen et al., 2009; Kerminen et al., 2005), and
atmospheric aerosol particles are thereby able to influence our climate indirectly, in addition to interacting directly with
incoming solar radiation. Though atmospheric aerosol particles provide the single largest cooling effect on our climate, they
are also connected with the greatest uncertainty in climate change projections (IPCC, 2013). Part of this uncertainty is caused
by limited knowledge about the aerosol precursor molecules.


Oxidation products of certain volatile organic compounds (VOCs) participate in both the formation of new particles

(Donahue et al., 2013; Schobesberger et al., 2013; Kulmala et al., 2014; Riccobono et al., 2014; Kirkby et al., 2016; Tröstl et

al., 2016) and growth of existing particles via gas-to-particle condensation (Riipinen et al., 2012; Ehn et al., 2014; Bianchi et

al., 2019). Globally, and especially in forested regions, the majority of these organic compounds originate from terrestrial

vegetation (Kanakidou et al., 2005; Jimenez et al., 2009). Thus, increases in the emissions of certain biogenic VOCs can lead

to enhanced formation of atmospheric aerosol particles and subsequently to a rise in CCN concentration (Kerminen et al.,

2012; Paasonen et al., 2013).

Biotic plant stress (i.e. stress caused to a plant by living species such as e.g. herbivores and pathogens) is known to

substantially alter both the rates of emission and spectrum of emitted VOCs (Holopainen and Gershenzon, 2010; Niinemets,

2010; Niinemets et al., 2013; Faiola and Taipale, 2020). For example, constitutively emitted isoprene, which is thought to

suppress the formation of new atmospheric aerosol particles (Kiendler-Scharr et al., 2009, 2012; Lee et al., 2016; McFiggans

et al., 2019; Heinritzi et al., 2020), is usually reduced in response to such stress (e.g. Brilli et al., 2009; Copolovici et al., 2014,

2017; Jiang et al., 2016), while the emissions of other VOCs are greatly induced. A large fraction of these stress-induced

compounds (e.g. monoterpenes, sesquiterpenes, 4,8-dimethyl-1,3,7-nonatriene (DMNT) and methyl salicylate) has a high

potential to produce and grow atmospheric aerosol particles (e.g. Mentel et al., 2013; Joutsensaari et al., 2015; Yli-Pirilä et al.,

2016; Ylisirniö et al., 2020), while other induced compounds (e.g. methanol and lipoxygenase oxidation products (LOX),

which mostly include $C_6$ aldehydes, alcohols and esters) are anticipated to suppress aerosol processes (e.g. Mentel et al., 2013).

While much attention has been given to representing constitutive emissions of VOCs in numerical models, VOC

emissions caused by stress, and in particular biotic stress, have been mostly excluded (Grote et al., 2013; Faiola and Taipale,

2020), despite the fact that biotic plant stress is largely ubiquitous. This is mainly due to a lack of measurements, combined

with the fact that variations in emission responses are highly stressor-specific (e.g. Holopainen and Gershenzon, 2010;

Niinemets, 2010; Faiola and Taipale, 2020). Thus, no consistent mechanism for the emissions of VOCs from plants under

stress has been proposed until now. Though the most extensively used biogenic emissions model, MEGAN (Guenther et al.,

2012), considers a stress emission category, the treatment is not quantitative. The emission factor for stress VOCs is, for

example, the same for all plant functional types and is supposed to represent a large range of different types of stresses.

Recently, Grote et al. (2019) proposed a new modelling framework for estimating emissions of VOCs induced by both biotic

and abiotic plant stresses, while Douma et al. (2019) developed a model to predict both the emissions and concentrations of

stress-induced VOCs, which was parameterized to simulate a gypsy moth infested black poplar canopy. Both are promising

tools, but in their current states, they exclude important storage emissions which are usually released upon wounding (e.g.

Blande et al., 2009; Faiola et al., 2018; Kari et al., 2019), and they do not fully (Grote et al., 2019) - or at all (Douma et al.,

2019) - consider how the constitutive emissions of isoprene are modulated. This is as such understandable considering that

emissions of isoprene might be either reduced (e.g. Brilli et al., 2009; Copolovici et al., 2017) or induced (e.g. Schaub et al.,

2010; Ye et al., 2019) in response to biotic plant stress, but nevertheless problematic since isoprene is globally the VOC that

is emitted in largest quantities (Guenther et al., 2012) and it is thought to suppress the formation of aerosol particles (see

above). Whilst Grote et al. (2019) assumed a linear response to the degree of damage, which has been shown not always to be

true, especially at severe stress levels (e.g. Niinemets et al., 2013; Jiang et al., 2016; Yli-Pirilä et al. 2016; Copolovici et al.,

2017; Faiola and Taipale, 2020), it is not obvious how the model results by Douma et al. (2019) depend on the degree of

damage, as they operate with "number of larvae" rather than a stand level of defoliation. Additionally, Grote et al. (2019) did

not account for an explicit dependency of the emissions on temperature, which is usually considered as one of the most

important environmental parameters for emissions of VOCs (e.g. Grote et al., 2013). Common for both studies is that they

only simulate rather short time scales (i.e. a few days).

Since measurements have clearly illustrated that biotic plant stress is able to significantly influence the amount and

size of formed atmospheric aerosol particles (Mentel et al., 2013; Joutsensaari et al., 2015; Yli-Pirilä et al., 2016; Faiola et al.,





2019, 2018) via perturbations in VOC emissions, there is an urgent need to quantify the atmospheric importance of biotic plant
stress. This need is amplified by the fact that insect outbreaks and fungal diseases generally are expected to increase in both
frequency and severity in the future (Cannon, 1998; Bale et al., 2002; Harrington et al., 2007; Pautasso et al., 2012; Boyd et
al., 2013). Unfortunately, such quantitative estimates are currently very scarce and connected with a large degree of
uncertainty. Bergström et al. (2014) used a regional chemical transport model to simulate the impact of *de novo* emissions,
induced by aphid infestation, on secondary organic aerosol (SOA) formation, and estimated that these induced emissions
currently account for 20-70 % of total biogenic SOA in northern and central European forests. Meanwhile, Joutsensaari et al.
(2015) calculated a local increase of up to 480 % in aerosol mass and 45 % in CCN concentration, when it was assumed that
10 % of the boreal forest area experienced stress which increased constitutive monoterpenes emission rates by an order of
magnitude. Using satellite observations, Joutsensaari et al. (2015) also found a 2-fold increase in aerosol optical depth over
Canadian pine forests during a bark beetle outbreak. To our knowledge, no one has previously considered the dynamics of
insect herbivory when simulating the emitted VOCs and produced and grown aerosols from stressed plants. Additionally, there
has so far been no attempts to measure nor model the impact of pathogenic infections on atmospheric aerosol processes.

We constructed a conceptual model to investigate the atmospheric impacts of biotic plant stresses. We used this model

to simulate formation and growth of atmospheric aerosol particles throughout a growing season in pure oak, poplar and birch
forest stands in stress-free conditions and under herbivory or fungal stress. By considering the dynamics of insect herbivory
and pathogenic infections in combination with seasonal changes in environmental parameters, our aim was to contribute to a
discussion about whether biotic plant stress perturbs atmospheric aerosol processes sufficiently to warrant their inclusion in
larger scale models.

## 2 Materials and methods

We constructed a model that includes modules for emissions of VOCs from stress-free and biotically stressed tree species (Sec.
2.4), boundary layer meteorology (Sec. 2.5), atmospheric chemistry (Sec. 2.6) and aerosol dynamics (Sec. 2.7). The calculated
canopy VOC emissions from biotically stressed trees depend on the dynamics of the biotic stressors of interest (Sec. 2.1 and
2.2) and changes in the leaf area index (Sec. 2.3).

### 2.1 Simulations of larval infestation dynamics

Whereas mountain birch (*Betula pubescens* var. *pumila*; former spp. *czerepanovii*, Fig. 1c) is the main host for autumnal moths
(*Epirrita autumnata*, Fig. 1g), European gypsy moth (*Lymantria dispar*, Fig. 1d) is one of the major defoliating insects feeding
on pedunculate oak (*Quercus robur*, Fig. 1a). The larval eggs of both moths hatch in spring synchronously with bud burst
(Kaitaniemi et al., 1997; Kaitaniemi and Ruohomäki, 1999; Spear, 2005; McManus et al., 1989). Both sexes have five larval
stages, though female gypsy moths have six. These stages are separated by periods of molting where the larvae do not feed. A
complete defoliation of vast areas can occur within 4-6 weeks by autumnal moth (Kaitaniemi and Ruohomäki, 1999) and
within 6-8 weeks by gypsy moth (McManus et al., 1989). Adults do not feed on leaves (Tammaru et al., 1996; Waring and
Townsend, 2009). For simulations of autumnal moth infested mountain birch, our model incorporates atmospheric and
ecological conditions observed at the Station for Measuring Ecosystem-Atmosphere Relations (SMEAR I, Fig. 1h), Värriö,
Eastern Finnish Lapland (e.g. Hari et al., 1994), due to the high data quality and availability, and since autumnal moth infested
mountain birch is common at the site (Hunter et al., 2014). In our simulations, bud burst occurs on 6th of June, the subsequent
full leaf state is attained on 10th of June (dates are based on long-term observations from the station), and senescence onsets
on 20th of August (Gill et al., 2015). We assumed that the larvae feed for five weeks, starting on 6th of June and they pupate on
11th of July (Kaitaniemi and Ruohomäki, 1999). The larvae dynamics (i.e. relative leaf consumption and time spent in each





larval stage) that is incorporated in our model is based on Lempa et al. (2004). For simulations of European gypsy moth infested
pedunculate oak, our model incorporates mainly atmospheric conditions observed at the Meteorological Observatory
Hohenpeißenberg (e.g. Birmili et al., 2003), rural southern Germany (Fig. 1h), since this station has been classified as a
representative measurement location for central Europe (Naja et al., 2003), where oak is a very common species. In our
simulations, bud burst occurs on 15[th] of May, the subsequent full leaf state is attained 20 days later on 4[th] of June, and
senescence onsets on 20[th] of September (Gill et al., 2015). Durations of the various developmental states of the larvae are based
on Zúbrik et al. (2007), which is also in agreement with Stoyenoff et al. (1994). Though the length of the larval state of female
larvae feeding on pedunculate oak is typically a few days longer than the total duration of the male larval state (e.g. Zúbrik et
al., 2007), we did not differentiate between the two genders, but utilised the length of the female larval state due to
simplification and since the female is the main consumer (Miller et al., 1991). We assumed that the larvae feed for 41 days
(Zúbrik et al., 2007), starting on 15[th] of May and they pupate on 25[th] of June. The relative leaf consumption within the different
larval stages is based on Kula et al. (2013). The ratio in relative leaf consumption between 4[th] and 5[th] instar is also within the
range that is reported by Stoyenoff et al. (1994) (where no other ratios were provided). We neglected periods of molting, as
those are typically in the order of less than one day (Ayres and MacLean, 1987). We assumed that either 30 % or 80 % of the
total leaf area in the forest stand was consumed by the end of the feeding period (Fig. 2a,b).

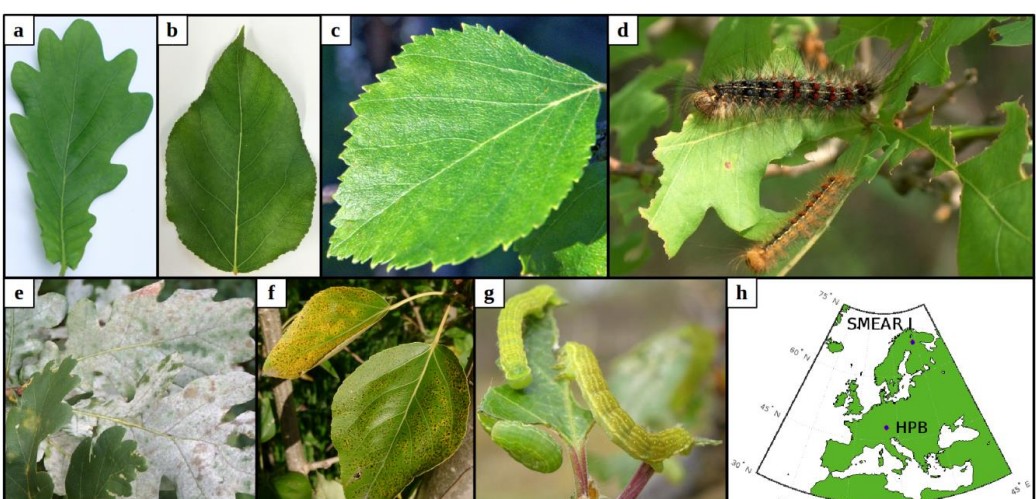

**Figure 1.** The plant species and biotic stresses we considered together with locations. **a-c**, non-infected leaves; pedunculate
oak (*Quercus robur*) **(a)**, balsam poplar (*Populus balsamifera* var. *suaveolens*) **(b)** and mountain birch (*Betula pubescens* var.
*pumila*; former spp. *czerepanovii*) **(c)**. **d-g**, fungal infected or moth infested leaves; pedunculate oak infested by European
gypsy moth (*Lymantria dispar*) **(d)** or infected by oak powdery mildew (*Erysiphe alphitoides*) **(e)**, poplar infected by rust
fungus (*Melampsora larici-populina*) **(f)** and mountain birch infested by autumnal moth (*Epirrita autumnata*) **(g)**. **h**, location
of the two sites that serve as boundary conditions in our simulations. Photo courtesy: a+c: Juho Aalto, b: Yifan Jiang, d+e:
Haruta Ovidiu, University of Oradea, Bugwood.org, f: Ülo Niinemets, g: Tero Klemola.

**2.2 Simulations of fungal infection dynamics**
Oak powdery mildew (*Erysiphe alphitoides*) is one of the main foliar diseases of pedunculate oak (*Quercus robur*) in Europe
(Desprez-Loustau et al., 2011). Among the Melampsora species, *Melampsora larici-populina* is the most widespread poplar
rust (Vialle et al., 2011). *M. larici-populina* has five morphologically and functionally different spore stages during its life





cycle, where the two first stages are retrieved on larch and the last three on poplar (Vialle et al., 2011). The pathogenic fungal
infections do not decrease the leaf area of their victim, but as they cover the leaf, they absorb nutrients from the cells of the
leaf (Glawe, 2008) and change the physiology of the leaf (e.g. Major et al., 2010; Voegele and Mendgen, 2003; El-Ghany et
al., 2009). The severity and spread of fungal infections depend largely on weather and growth conditions (e.g. Åhman, 1998;
Johansson and Alström, 2000; Covarelli et al., 2013), where especially high rainfall in the beginning of the summer greatly
enhances both the severity, but also the onset of infection (Covarelli et al., 2013; Pinon et al., 2006). The onset of attack by
oak powdery mildew is limited by its morphological development, hence the infection usually starts to appear between the end
of June and August (in France; Marçais et al., 2009; Marçais and Desprez-Loustau, 2014; Bert et al., 2016). *M. larici-populina*
has been observed to attack young poplar trees as early as June (in Italy, Covarelli et al., 2013), though generally the disease
emerges between July and September (in France and Italy, Gérard et al., 2006; Covarelli et al., 2013). In our simulations, we
assumed that both fungi started to infect their host on 1st of August. In the case of *M. larici-populina* we only simulated the
attack on poplar as the host (and not larch). *Populus balsamifera* var. *suaveolens* was chosen as the poplar species due to the
availability of suitable published VOC emissions measurements. Based on Bert et al. (2016), we assumed that the severity of
infection increases linearly with time, starting on 1st of August and ending on 20th of September. We assumed that either 30 %
or 80 % of the total leaf area in the forest stands was covered by fungi by the end of the growing season (Fig. 2c,d). For these
simulations, our model incorporates the same atmospheric conditions as for simulations of gypsy moth infested oak. Poplar
bud burst occurs on 20th of April in our model (Tripathi et al., 2016) and we assumed the same timing of senescence as for
simulations of oak (Gill et al., 2015; Tripathi et al., 2016).



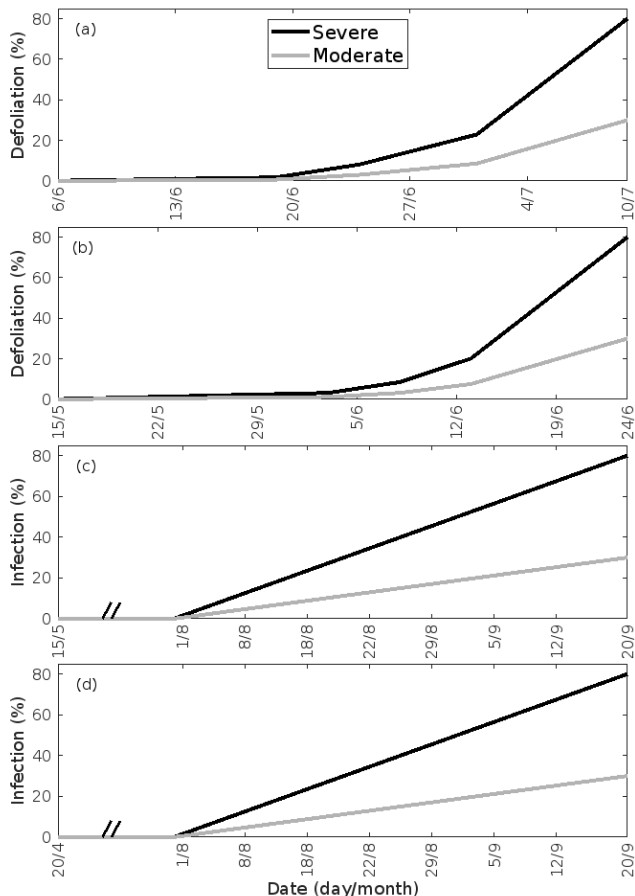


**Figure 2.** Infection dynamics. **(a)** birch infested with autumnal moth larvae, **(b)** oak infested with European gypsy moth larvae,
**(c)** oak infected by oak powdery mildew and **(d)** poplar infected by rust fungus. The infection dynamics of oak powdery
mildew and rust fungus is assumed to be similar, but the duration of growth of the two tree species is different. The dynamics
are specific to the locations of Lapland **(a)** and central Europe **(b-d)**. 30 % (moderate infection scenario) or 80 % (severe
infection scenario) of the total leaf area in the forest stands is assumed to be consumed by the end of the feeding period or
infected by fungi by the onset of senescence. Note the different time axes.

**2.3 Treatment of the leaf area index**
Low soil temperatures usually prevent growth of a second leaf flush in birch trees (Aphalo et al., 2006). Thus, we assumed
that the leaf area index (LAI) of a non-infested mountain birch stand in Lapland increases linearly from $0 – 2.0$ m$^2$ m$^{-2}$
(Dahlberg et al., 2004) in the time period 6th-10th of June (bud burst to full leaf), and stays constant until the onset of senescence
(Fig. 3a). The LAI of an infested stand decreases proportionally with the level of defoliation (Fig. 3a). Refoliation only occurs
in totally, or near-totally, defoliated mountain birch trees (Kaitaniemi et al., 1997). Hence, we assumed that the LAI of a
heavily defoliated stand resumes to 70 % of the original LAI within three weeks of defoliation (personal communication with
adj. Prof. Dr. Pekka Kaitaniemi, Fig. 3a). Poplar trees produce leaves throughout the season, and we therefore assumed that
the LAI of poplar stands increases quadratically from 20th of April (bud burst) until 15th of August (Fig. 3c, Tripathi et al.,
2016). Oak, on the other hand, usually only produces one significant leaf flush, hence we assumed that the LAI of oak stands





increases with a sigmoid shape from 15[th] of May (bud burst) until 4[th] of June (full leaf state attained) (Fig. 3b, Oláh et al.,
2012). In our simulations, the maximum LAI of poplar and oak is 5.0 $m^2$ $m^{-2}$. The LAI of a gypsy moth infested oak stand
decreases proportionally with the level of defoliation (Fig. 3b). The fungal infections do not decrease the leaf area of their host,
nor do they prevent the tree from producing multiple flushes of leaves (Marçais and Desprez-Loustau, 2014). The LAI might,
however, in reality be less in an infected stand than in a non-infected stand, though this depends highly on the specific
genotypes and their individual fungal resistance (Verlinden et al., 2013; Shifflett et al., 2016), but naturally also on the timing
of infection. Since most summer leaves already appear before the onset of infection, we did not assume a decrease in LAI.
Severe powdery mildew infection (>50 %) has been shown to greatly reduce the infected leaf lifespan (Hajji et al., 2009). The
median time before shedding of deformed oak leaves has been estimated to be 10-31 days (Hajji et al., 2009). In our scenario
of a heavily infected stand (80 % of the stand leaf area is infected by the end of the season), an infection level of 50 % is
reached on 1[st] of September. Since senescence is assumed to onset on 20[th] of September, we excluded an earlier shedding of
leaves. Hence, in our simulations of fungal infections, we assumed that the LAI is the same as in a non-infected stand (Fig.
3b,c).

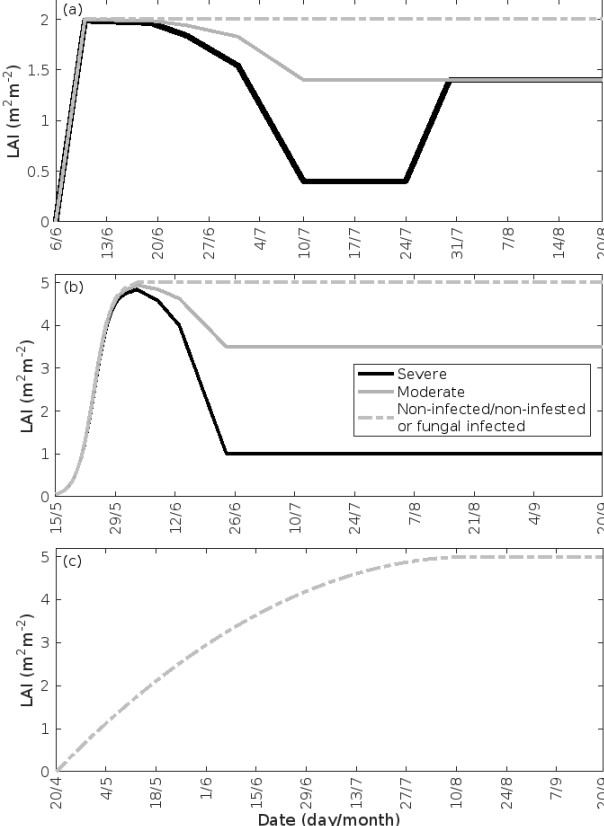

**Figure 3.** Leaf area index throughout the growing season in infected and non-infected forest stands. **(a)** mountain birch, **(b)**
oak and **(c)** poplar. 30 % (moderate infection scenario) or 80 % (severe infection scenario) of the total leaf area in the forest
stands is assumed to be consumed by the end of the feeding period in simulations of herbivory, while fungal infections do not
decrease the leaf area. Note the different time axes.



### 2.4 Plant emissions of volatile organic compounds

The plant emissions ($F_i$) of individual VOCs ($i$) from various pure stands were computed as:

$$F_i = \varepsilon_i \times \text{LAI} \times \gamma_L \times \gamma_T \tag{1}$$

where $\varepsilon_i$ is the emission rate of $i$ at standard conditions (25 ºC, 1000 µmol m$^{-2}$ s$^{-1}$), LAI is the leaf area index and treated as mentioned in Sec. 2.3, and $\gamma_L$ and $\gamma_T$ are the activity factors that account for changes in light and temperature from standard conditions. This expression is adopted from Guenther et al. (2012), Eq. 1-2, when we assume that the soil moisture and ambient $CO_2$ concentration in our stands are optimal. Generally, we excluded the effect of leaf age on the emissions of VOCs, since we do not know the effect of leaf age on stress-induced emissions. However, we also tested whether the impact of leaf maturity would be able to change the conclusions of our study when making certain assumptions about the leaf age effect. The treatment of the leaf age effects and the results of these tests are presented in Appendix A and discussed in Sec. 3.2.

Similarly to e.g. Simpson et al. (1999; 2012) and Bergström et al. (2014), we utilised a simple non-canopy approach, where ambient and leaf temperature are assumed to equal, and where the use of branch-level emission factors accounts for the canopy shading effect (Guenther et al., 1994). We utilised emission rates, reported as a function of the degree of damage, from Copolovici et al. (2017; 2014), Jiang et al. (2016), and Yli-Pirilä et al. (2016). The emission rates reported by Copolovici et al. (2017; 2014) and Jiang et al. (2016) were all retrieved by leaf-level measurements, hence we decreased the reported emission rates by a factor of 0.57, since branch-level emission factors for light sensitive emissions are typically a factor of 1.75 smaller than the corresponding leaf-level values (Simpson et al., 1999; 2012). We did not decrease the emission rates reported by Yli-Pirilä et al. (2016), since these were based on measurements of the whole plant. Instead, the emission rates from mountain birch seedlings (Yli-Pirilä et al., 2016) were upscaled in order to represent the emissions from mature trees, assuming a leaf mass area of 75 g m$^{-2}$ for leaves growing on mature mountain birch trees (Riipi et al., 2005). Since emission response measurements are usually stopped, at a maximum, a few days after the herbivore activity has ceased, we assumed that the effect of stress on the emissions of VOC stops the day that the larvae pupate (e.g. Yli-Pirilä et al., 2016), in order to not overestimate the impact of the stress. The light and temperature dependent emission activity factors are computed using Eq. 2 in Guenther (1997), since none of the considered broadleaved species poses storages and birch has specifically been shown to only emit *de novo* (Ghirardo et al., 2010). Similarly to Guenther et al. (2012), we assumed that stress-induced emissions are controlled by light and temperature in a similar way as constitutive emissions, thus the used emission rates from the literature were standardised according to Eq. 1. Copolovici et al. (2014) and Jiang et al. (2016) have also shown that the emissions of isoprene from oak powdery mildew and rust infected oaks and poplars have the same response to light as control plants. Copolovici et al. (2014) additionally demonstrated that the emissions of monoterpenes from oak powdery mildew infected oak depend strongly on light, even though the majority of emitted monoterpenes were not e.g. ocimene and linalool, which are known to be light dependent (Niinemets et al., 2002; Arimura et al., 2008). LOX (lipoxygenase pathway volatile) compounds, on the other hand, are released shortly after damage of leaf tissue, independent of the light conditions (Arimura et al., 2008). LOX compounds do not contribute to the formation and growth processes of atmospheric aerosol particles, but they were included in the model in order to illustrate the changes in the atmospheric concentrations of LOX as a function of stress severity and stress type, and for evaluating the reliability of our modelling results, since LOX, in reality, affect the atmospheric concentration of OH, which was constrained in the model. The equations, which we used in the model for linking the emission factors to the severity of stress, are provided in Table 1 together with the parameters needed for the equations.

**Table 1.** Equations to calculate the emission factors ($\varepsilon_{i,D}$, in unit nmol m$^{-2}$ s$^{-1}$), as a function of the degree of stress ($D$), together with the parameters needed for the equations. The equations are valid for infection levels ranging from 0 % to 80 % unless otherwise stated. The emission factors, listed for oak and poplar in the table, have not been downscaled (by a factor of 0.57), but the emission factors for mountain birch, listed here, have been upscaled in order to represent the emissions from mature trees. Thus, LMA$_f$ is the fraction of the leaf mass area of leaves growing on mature mountain birch / growing on mountain



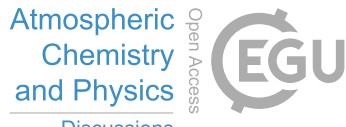

birch seedlings. ISO = isoprene, MT = monoterpenes, MeSa = methyl salicylate, LOX = lipoxygenase pathway volatile
compounds, DMNT = 4,8-dimethyl-1,3,7-nonatriene, MeOH = methanol, SQT = sesquiterpenes, α-Eud = α-Eudesmol.

**Infestation of pedunculate oak (*Quercus robur*) by European gypsy moth (*Lymantria dispar*) based on Copolovici et al. (2017).**

| VOC | $\varepsilon_{i,D}$ (nmol m$^{-2}$ s$^{-1}$) | $\varepsilon_{i,0}$ (nmol m$^{-2}$ s$^{-1}$) | A (nmol m$^{-2}$ s$^{-1}$) | B | C (nmol m$^{-2}$ s$^{-1}$) | |
|---|---|---|---|---|---|---|
| ISO | $\varepsilon_{i,0} + \frac{D \times A}{B+D}, \quad 0 \leq D \leq 60$ <br> $\varepsilon_{i,60} \times 0.5, \quad 60 < D \leq 80$ | 30.26 | -47.40 | 34.48 | | |
| MT | $\varepsilon_{i,0} + \frac{D \times A}{B + D}$ | $4.0 \cdot 10^{-2}$ | 9.22 | 33.42 | | |
| MeSa | $D \times A$ | | $3.5 \cdot 10^{-3}$ | | | |
| LOX | $A \times D^2 + C \times D$ | | $1.1 \cdot 10^{-3}$ | | $4.7 \cdot 10^{-2}$ | |
| DMNT | $A \times D^2 + C \times D$ | | $1.0 \cdot 10^{-5}$ | | $1.3 \cdot 10^{-3}$ | |

**Infection of pedunculate oak (*Quercus robur*) by oak powdery mildew (*Erysiphe alphitoides*) based on Copolovici et al. (2014)**

| VOC | $\varepsilon_{i,D}$ (nmol m$^{-2}$ s$^{-1}$) | $\varepsilon_{i,0}$ (nmol m$^{-2}$ s$^{-1}$) | A (nmol m$^{-2}$ s$^{-1}$) | B (nmol m$^{-2}$ s$^{-1}$) | C (nmol m$^{-2}$ s$^{-1}$) | |
|---|---|---|---|---|---|---|
| ISO | $\varepsilon_{i,0} + A \times D + B \times D^2 + C \times D^3$ | 10.6 | -0.244 | $3.69 \cdot 10^{-3}$ | $-2.05 \cdot 10^{-5}$ | |
| MT | $\varepsilon_{i,0} + A \times D + B \times D^2 + C \times D^3$ | $4.0 \cdot 10^{-2}$ | $8.7 \cdot 10^{-3}$ | $-7.1 \cdot 10^{-5}$ | $3.7 \cdot 10^{-7}$ | |
| MeSa | $0, \quad D = 0$ <br> $A \times \frac{\varepsilon_{MT,D}}{\varepsilon_{MT,60}}, 0 < D \leq 80$ | | 0.437 | | | |
| LOX | $A + B \times D$ | | $2.13 \cdot 10^{-3}$ | $6.24 \cdot 10^{-3}$ | | |

**Infection of balsam poplar (*Populus balsamifera* var. *suaveolens*) by poplar rust (*Melampsora larici-populina*) based on Jiang et al. (2016)**

| VOC | $\varepsilon_{i,D}$ (nmol m$^{-2}$ s$^{-1}$) | $\varepsilon_{i,0}$ (nmol m$^{-2}$ s$^{-1}$) | A (nmol m$^{-2}$ s$^{-1}$) | B (nmol m$^{-2}$ s$^{-1}$) | C | E (nmol m$^{-2}$ s$^{-1}$) |
|---|---|---|---|---|---|---|
| ISO | $A + \frac{B}{C + D}$ | | 12.3 | 366.8 | 4.98 | |
| MT | $0.0625, D = 0$ <br> $A + B \times D + E \times D^2, 0 < D \leq 80$ | | 0.112 | $1.84 \cdot 10^{-3}$ | | $1.5 \cdot 10^{-4}$ |
| MeSa | $A \times D + B \times D^2 + E \times D^3$ | | $6.32 \cdot 10^{-3}$ | $-8.6 \cdot 10^{-5}$ | | $5.75 \cdot 10^{-7}$ |
| LOX | $0.4814, D = 0$ | | 2.51 | $-2.51 \cdot 10^{-2}$ | $0.76, 0 < D < 30$ | $3.25 \cdot 10^{-3}$ |





| | | | | | | |
|---|---|---|---|---|---|---|
| | $(A + B \times D + E \times D^2) \times C,\ 0 < D \leq 80$ | | | | 0.85, $30 \leq D < 60$<br>0.93, $60 \leq D \leq 80$ | |
| DMNT | $\varepsilon_{\text{DMNT},60} \times \frac{\varepsilon_{\text{MeSa},D}}{\varepsilon_{\text{MeSa},60}},\quad 0 \leq D < 60$<br>$\varepsilon_{\text{MeSa},D} \times C,\quad 60 \leq D \leq 80$ | | | | 0.36 | |
| MeOH | $\varepsilon_{i,0} + A \times D + B \times D^2$ | 16.9 | -0.338 | $1.36 \cdot 10^{-2}$ | | |
| SQT | $\varepsilon_{\text{SQT},60} \times \frac{\varepsilon_{\text{MeSa},D}}{\varepsilon_{\text{MeSa},60}},\quad 0 \leq D < 60$<br>$\varepsilon_{\text{MeSa},D} \times C,\quad 60 \leq D \leq 80$ | | | | 2.414 | |
| α-Eud | $\varepsilon_{\alpha\text{-Eud},60} \times \frac{\varepsilon_{\text{MeSa},D}}{\varepsilon_{\text{MeSa},60}},\quad 0 \leq D < 60$<br>$\varepsilon_{\text{MeSa},D} \times C,\quad 60 \leq D \leq 80$ | | | | 0.397 | |
| Infestation of mountain birch (*Betula pubescens* var. *pumila*) by autumnal moth (*Epirrita autumnata*) based on Yli-Pirilä et al. (2016) | | | | | | |
| VOC | $\varepsilon_{i,D}$ (nmol m$^{-2}$ s$^{-1}$) | A | B | C | E<br>(nmol m$^{-2}$ s$^{-1}$) | LMA$_f$ |
| MT | $\left(A + \dfrac{D \times B}{\sqrt{1 + \dfrac{B^2 \times D^2}{C^2}}}\right) \times E \times \text{LMA}_f$ | $7.65 \cdot 10^{-2}$ | $9.33 \cdot 10^{-3}$ | 0.2146 | 0.769 | 2.23 |
| LOX | $(A \times D + B) \times E \times \text{LMA}_f$ | $6.325 \cdot 10^{-3}$ | $4.868 \cdot 10^{-2}$ | | 0.8 | 2.23 |
| DMNT | $\left(A + \dfrac{D \times B}{\sqrt{1 + \dfrac{B^2 \times D^2}{C^2}}}\right) \times E \times \text{LMA}_f$ | $7.11 \cdot 10^{-4}$ | $3.39 \cdot 10^{-4}$ | $8.63 \cdot 10^{-3}$ | 0.769 | 2.23 |
| SQT | $\dfrac{\varepsilon_{\text{MT},D}}{3}$ | | | | | |


## 2.5 Meteorological conditions

The daily maximum radiation during the entire growing season was fixed to 1000 μmol m$^{-2}$ s$^{-1}$ (Table 2), which corresponds
to the average maximum photosynthetic photon flux density (PPFD) observed at the SMEAR I station during the growing
seasons of 2015-2017 (Aalto et al., 2019). The daily pattern of PPFD then follows the solar zenith angle. For simulations of
mountain birch, we utilised the maximum and minimum mean temperatures on every day in the growing season during 2015-
2017 observed at 9 m at the SMEAR I station (Aalto et al., 2019). The daily maximum and minimum temperatures ranged
from 9.8 to 19.6 °C, and from 2.0 to 11.3 °C, respectively, in the time period of interest (6$^{th}$ of June - 20$^{th}$ of August, Fig. 4a).
For simulations of oak and poplar, we utilised the maximum and minimum temperatures for southern Germany averaged over
the past three decades (data obtained via https://www.currentresults.com/Weather/Germany/average-annual-
temperatures.php). This was done due to availability and restriction of data obtained at the Hohenpeißenberg Meteorological
Observatory and since our aim was not as such to simulate the atmospheric impact at Hohenpeißenberg, but instead at any
relevant location, i.e. where oaks and poplars, including the biotic stresses of interests, are common. The monthly averaged
daily maximum and minimum temperatures ranged from 15 to 26 °C, and from 6 to 16 °C, respectively, in the time period of
interest (April - September, Fig. 4b). For simplicity, the daily temperature pattern followed that of the solar zenith angle with





a forward shift of 1 h. The default daytime mixing length was kept constant to a value of 700 m (simulations of mountain
birch) and 2000 m (simulations of oak and poplar) above ground level (Seidel et al., 2012) (Table 2).

**Table 2.** Model inputs. Representative summer time conditions in rural central Europe (here indicated by HPB: the
Hohenpeißenberg Meteorological Observatory) and Lapland (here indicated by the SMEAR I station), used as default model
input. The conditions are chosen such that they are realistic and representative, but they do not inhibit the formation of new
particles. The concentrations of OH and sulfuric acid are provided as daily maxima in the table, but their concentrations
decrease as a function of the decrease in solar light in the simulations. The concentration of ozone is kept constant throughout
the simulations, while the condensation sink (CS) takes into account the relative importance of sulfuric acid and oxidised
organic compounds on the CS (Eq. 5; Peräkylä et al., 2020). The condensation sink for sulfuric acid ($CS_{SA}$) is kept constant
throughout the simulations. BHL is the planetary boundary layer height. The photosynthetic photon flux density (PPFD) is
provided as the daily maximum in the table, but the daily pattern of PPFD follows the solar zenith angle in the model.

|  | HPB | SMEAR I |
|---|---|---|
| $[O_3]$ (ppb) | 45 | 30 |
| $[OH]_{max}$ (molec cm$^{-3}$) | $6 \cdot 10^6$ | $8 \cdot 10^5$ |
| $[H_2SO_4]_{max}$ (molec cm$^{-3}$) | $1 \cdot 10^7$ | $2.5 \cdot 10^6$ |
| $CS_{SA}$ (s$^{-1}$) | $2.5 \cdot 10^{-3}$ | $7 \cdot 10^{-4}$ |
| BLH (m) | 2000 | 700 |
| $PPFD_{max}$ (µmol m$^{-2}$ s$^{-1}$) | 1000 | |


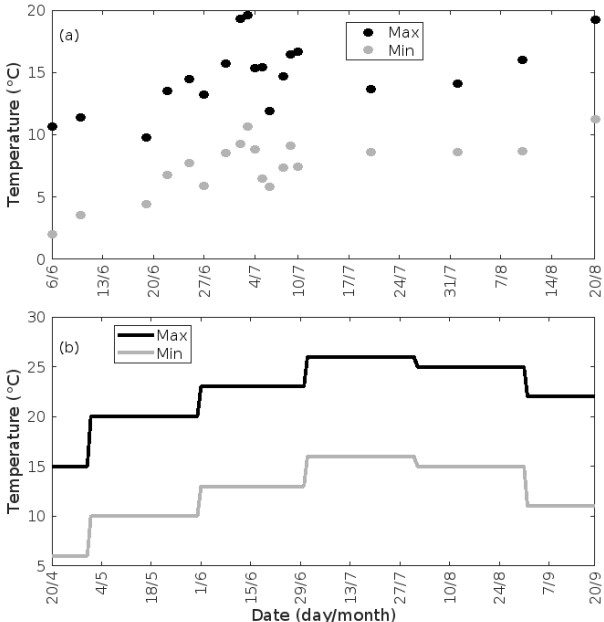


**Figure 4.** Daily maximum and minimum temperatures throughout the growing season at **(a)** SMEAR I, Lapland, and **(b)**
Southern Germany, used as default model input. Note the different time axes.





**2.6 Atmospheric gas phase chemistry**

Similarly to previous atmospheric modelling studies of herbivory (Bergström et al., 2014; Douma et al., 2019), we constrained the concentrations of atmospheric oxidants within the model, since it is unreasonable to assume that they can be accurately predicted. Many studies do, for example, report a large (up to at least 89 %) missing sink of OH, especially in forested areas (e.g. Di Carlo et al., 2004; Sinha et al., 2010; Mogensen et al., 2011, 2015; Nölscher et al., 2012, 2016; Zannoni et al., 2016; Praplan et al., 2019) and studies above the Amazonian rainforest, furthermore, indicate that isoprene can recycle OH with a varying efficiency of 40-120 % (Lelieveld et al., 2008; Taraborrelli et al., 2012). In our simulations, the default daily maximum concentration of OH is therefore fixed to $6 \cdot 10^6$ molec cm$^{-3}$ (Petäjä et al., 2009) and $8 \cdot 10^5$ molec cm$^{-3}$ (calculated using observed summertime UVB radiation from the SMEAR I station and the proxy presented by Petäjä et al. (2009)) for simulations of Hohenpeißenberg and Lapland, respectively (Table 2). The daily pattern of the OH concentration then follows the solar zenith angle. The concentration of ozone is kept constant to a value of 45 ppb (Naja et al., 2003) and 30 ppb (Ruuskanen et al., 2003) for simulations of oak and poplar (Hohenpeißenberg conditions) and mountain birch (SMEAR I conditions), respectively (Table 2). In reality, the atmospheric oxidant concentration can, however, decrease or increase depending on changes in the concentrations of individual specific VOCs (Table 3). The impact of changing oxidation concentrations on our simulation results was therefore also tested (Sec. 3.2, Fig. 11a-b,f-g,k-l,p-q).

The only source of sulfuric acid (H$_2$SO$_4$), in our model, is the reaction between OH and SO$_2$, while the only sink is the condensation sink. The concentration of SO$_2$ is chosen such that the default daytime maximum concentration of H$_2$SO$_4$ is $1 \cdot 10^7$ molec cm$^{-3}$ in Hohenpeißenberg (Petäjä et al., 2009; Birmili et al., 2003) and $2.5 \cdot 10^6$ molec cm$^{-3}$ in Lapland (Kyrö et al., 2014) (Table 2). The size distribution of the pre-existing particle population is kept fixed during the simulations, so the number concentration of pre-existing particles is defined by the condensation sink (CS). The overall value of CS is related to that of sulfuric acid (CS$_{SA}$) via Eq. 5 that takes into account the relative importance of sulfuric acid and oxidised organic compounds on CS (Peräkylä et al., 2020). The condensation sink of sulfuric acid is kept constant to a value of $2.5 \cdot 10^{-3}$ s$^{-1}$ (in rural southern Germany) and $7 \cdot 10^{-4}$ s$^{-1}$ (in Lapland; Dal Maso et al., 2007; Vana et al., 2016) during our simulations (Table 2).

We included reactions for the atmospheric oxidation of SO$_2$ and the emitted VOCs (Appendix B). Certain VOCs, and especially VOCs with endocyclic double bonds, can form HOM (highly oxygenated organic molecules) upon oxidation by, in particular O$_3$, but also OH and NO$_3$ (e.g. Ehn et al., 2012, 2014; Jokinen et al., 2015; Berndt et al., 2016; Bianchi et al., 2019; Zhao et al., 2020). HOM have been found to be a major component of secondary organic aerosol (e.g. Ehn et al., 2014; Mutzel et al., 2015). HOM yields are specific to individual molecules and isomers and most yields have not been investigated for the exact compounds, which are emitted from the tree species considered in this study. Thus, the yields applied for the production of HOM in the model (Appendix B) are connected with a large degree of uncertainty. The influence of changing HOM yields on our results was therefore also investigated (Sec. 3.2, Fig. 11e,j,o,t). Formation of oxygenated organics from oxidation of sesquiterpenes and methyl salicylate are also included (Appendix B). The sum of all organic compounds, which contribute to aerosol processes, is referred to as "OxOrg".





**Table 3.** Common changes in the atmospheric concentrations of volatile organic compounds (VOC) and OH during biotic
plant stress. LOX are lipoxygenase pathway volatile compounds.

| | Change in VOC concentration | Change in OH concentration | Reference |
|---|---|---|---|
| Stress↑ | [LOX]↑<br>[methyl salicylate]↑<br>[methanol]↑<br>[monoterpenes]↑ | [OH]↓ | Mentel et al. (2013)<br>Calvert et al. (2000)<br>Atkinson et al. (1992)<br>Hakola et al. (1994) |
| | [sesquiterpenes]↑ | [OH]↑ | Atkinson and Arey (2003)<br>Winterhalter et al. (2009) |
| | [isoprene]↓ | [OH]?, but most probably [OH]↑ | Lelieveld et al. (2008)<br>Taraborrelli et al. (2012)<br>Wells et al. (2020) |


### 2.7 Calculation of the formation and growth of secondary organic aerosol particles

The clustering and activation of new particles are expressed by a formation rate of neutral 2 nm sized clusters, $J_2$ (cm$^{-3}$ s$^{-1}$),
which is computed by Eq. 20, using coefficients ($\alpha_{1-3}$) from Table 3, both found in Paasonen et al. (2010):
$$J_2 = \alpha_1 \times [H_2SO_4]^2 + \alpha_2 \times [H_2SO_4][OxOrg] + \alpha_3 \times [OxOrg]^2 \qquad (2)$$
It is here assumed that new particles are formed via heteromolecular homogeneous nucleation between sulfuric acid and
oxidised organic compounds (OxOrg) as well as via homogeneous nucleation of sulfuric acid and OxOrg alone. For
simplification, we only operated with one growing aerosol mode and therefore included a unit-less correction term (KK),
which determines how large a fraction of the activated clusters reaches the growing mode (Kerminen and Kulmala, 2002):
$$KK = exp\big(\eta \times [1/D_p - 1/D_{clus}]\big) \qquad (3)$$
where $D_p$ and $D_{clus}$ are the diameters of the growing mode and clusters, respectively, and $D_{clus} = 2$ nm as stated above. Further,
$\eta$ (nm) is (Eq. 11-12 and Table 1 in Kerminen and Kulmala, 2002):
$$\eta = 1830 \; nm^2 \cdot s \cdot h^{-1} \times CS/GR \qquad (4)$$
and we account for the the fact that the condensation sink (CS) is compound specific and less for oxidised organic
compounds than for sulfuric acid (CS$_{SA}$) (Peräkylä et al., 2020):
$$CS = CS_{SA} \times \frac{[H_2SO_4]}{[H_2SO_4]+[OxOrg]} + CS_{SA} \times 0.5 \times \frac{[OxOrg]}{[H_2SO_4]+[OxOrg]} \qquad (5)$$
The condensational particle diameter growth rate (GR, nm h$^{-1}$) of newly formed 2-3 nm particles is calculated according to
Nieminen et al. (2010):
$$GR_{2-3 \, nm} = 0.5 \; nm \cdot h^{-1} \times CC \times 10^{-7} \; cm^3 \qquad (6)$$
where CC is the concentration of condensable vapours which we assumed to be the sum of sulfuric acid and OxOrg. In addition,
we assumed that the molar mass of OxOrg are 3.5 times larger than that of sulfuric acid (Ehn et al., 2014), hence:
$$CC = [H_2SO_4] + [OxOrg] \times 3.5^{1/3} \qquad (7)$$
It is a complex matter to model nanoparticle growth, especially in forested environments, since thousands of individual
molecules with different vapour pressures contribute to the growth, but particle growth rates have been observed to be strongly
size-dependent in the field (Hirsikko et al., 2005; Yli-Juuti et al., 2011; Häkkinen et al., 2013). Hence, we accounted for this
size-dependency by enhancing the growth rates of particles larger than 3 nm, as according to Hirsikko et al. (2005) and Yli-
Juuti et al. (2011):
$$GR_{3-7 \, nm} = 2 \times GR_{2-3 \, nm} \qquad (8)$$
$$GR_{>7 \, nm} = 2.3 \times GR_{2-3 \, nm} \qquad (9)$$



The increase in the diameter of the growing mode ($D_p$) is defined by the growth rate:
$\Delta D_p/\Delta t = \text{GR}/3600 \text{ s} \cdot \text{h}^{-1}$                                                                                   (10)
while the increase in the number of new particles ($N_p$, cm$^{-3}$) is determined by the formation of new particles which reaches
the growing mode and the coagulation of particles in the growing mode by:
$\Delta N_p/\Delta t = J_2 \times \text{KK} - \text{CoagS} \times N_p$                                                                         (11)
where the coagulation sink (CoagS, s$^{-1}$) is calculated according to (Lehtinen et al., 2007):
$\text{CoagS} = \text{CS} \times \left(0.71 \text{ nm}/D_p\right)^{1.6}$                                                                    (12)

## 3 Results and discussion

### 3.1 Simulations of biotically stressed and non-infected forest stands throughout a growing season

Simulation results of one full growing season from various non-infected, and moderately and severely infected forest stands
are presented in Figs. 5-9. In the figures, emissions, concentrations, the isoprene-to-monoterpenes carbon concentration ratio
(R = [isoprene C]/[monoterpenes C] = 0.5·[isoprene]/[monoterpenes]), the formation and growth rates and number of newly
formed particles are expressed as median values during 10:00-16:00 local time, while the particle diameter of the growing
mode is provided as the daily maximum.

### 3.1.1 Canopy emissions of VOCs

The emissions of VOCs (Figs. 5a,c,e,g, 7a,b, 8a,b, 9a-c) change throughout the season due to variations in temperature, light,
LAI and infection severity. The impact of leaf maturity development on emission predictions are presented in Appendix A and
its atmospheric relevance discussed in Sec. 3.2. Canopy emissions of VOCs are highly different from non-infected and infected
forest stands, due to plant stress responses, but also due to a decrease in LAI in case of larval infestations (Figs. 5a,c,e,g, 9a-
c). Constitutive isoprene emitters (i.e. oak and poplar) decrease their emissions of isoprene significantly during the episodes
of biotic stress, and a stronger reduction is observed as a function of an increase in stress severity (Figs. 5a, 7a, 8a). All
investigated stresses cause the plants to induce their emissions of monoterpenes greatly (Figs. 5c, 7b, 8b, 9a). The induction
in the emissions of monoterpenes increases as a function of stress severity per unit leaf area, but the LAI simultaneously
decreases in case of larval infestations, which result in smaller canopy scale emissions from severely defoliated stands
compared to moderately stressed stands (Figs. 5c, 9a). Also the emissions of compounds, and groups of compounds, such as
methyl salicylate, LOX, methanol, DMNT, sesquiterpenes and oxygenated sesquiterpenes are significantly induced as a result
of biotic plant stress (Figs. 5e,g, 7a,b, 8a,b, 9b,c). These compounds, together with monoterpenes, are in most cases not emitted
constitutively at all or only in very small abundances. Though the emissions of all induced VOCs increase as a function of the
degree of damage, the responses to the level of stress severity are not necessarily the same for all VOCs and all individual
stresses. This difference can, for example, be seen in the emissions of monoterpenes (Fig. 5c) and LOX compounds (Fig. 5e)
from gypsy moth infested oak forests, which do not peak at the same time in the season.

### 3.1.2 Ambient concentrations of VOCs and OxOrg

Since the concentrations of OH and O$_3$ are constrained within the simulations, the VOC emission patterns are reflected in the
concentration patterns (Figs. 5b,d,f,h, 7c-e, 8c-e, 9e-g). All VOCs, except LOX and methanol, contribute to the formation of
OxOrg (Figs. 6a, 7c, 8c, 9d), but the contributions from oxidation of the individual VOCs, or groups of VOCs, vary between





the various stress cases and infection severity levels due to differences in emission rates and OxOrg forming yields (Appendix
B). For example, in herbivory infested stands, OxOrg originating from monoterpenes make up by far the largest fraction of
total OxOrg. This is mainly due to the fact that the induced emissions of monoterpenes are significantly higher than the
emissions of other VOCs which contribute to OxOrg formation (Figs. 5c,g, 9a,c). In case of oak powdery mildew infected oak,
HOM from monoterpenes and oxygenated organics from methyl salicylate reactions contribute about evenly to the total OxOrg.
The reason for this is that the canopy emissions of these VOCs are roughly similar (Fig. 7a,b), the OxOrg yield of methyl
salicylate is significantly higher than that of monoterpenes, but oxidation of methyl salicylate is correspondingly slower
(Appendix B). The main contributor to the total OxOrg in poplar rust infected poplar stands is sesquiterpenes. Contributions
from methyl salicylate, DMNT and α-Eudesmol to total OxOrg are individually rather small, but together, they are close to
matching the contribution from monoterpenes. When emissions of sesquiterpenes are omitted from simulations of a severely
rust infected poplar stand, the concentration of OxOrg decreases with ~46 %, while, in comparison, the concentration of OxOrg
decreases with ~30 % if the emissions of monoterpenes are instead excluded.
In the simulations, the daily median (10:00-16:00) ambient concentration of OxOrg is at maximum $\sim 4.2 \times 10^7 \text{ cm}^{-3}$
in a gypsy moth infested oak stand (Fig. 6a), $\sim 1.1 \times 10^7 \text{ cm}^{-3}$ in an oak powdery mildew infected oak stand (Fig. 7c), $\sim$
$4.2 \times 10^7 \text{ cm}^{-3}$ in a rust infected poplar stand (Fig. 8c), and $\sim 3.3 \times 10^7 \text{ cm}^{-3}$ in an autumnal moth infested mountain birch
stand (Fig. 9d). The ambient concentration of OxOrg is much higher in gypsy moth infested oak stands, than in a non-infested
oak stand, during the period of stress (Fig. 6a). When the period of feeding has been concluded, the concentration of OxOrg is
higher in non-infested oak stands than in stands that have been exposed to stress due to a higher LAI. However, the
concentration of OxOrg is then only ~15-20 % of the OxOrg concentration during the period of stress and it is almost
exclusively composed of HOM originating from isoprene. The concentration of OxOrg increases as a function of fungal
infection severity, and in our simulations the concentration of OxOrg is higher in fungally infected stands by a factor of up to
~6.9 (oak powdery mildew infected oak stands, Fig. 7c) and ~3.3 (poplar rust infected poplar stands, Fig. 8c). Since the
investigated poplar species is a great constitutive isoprene emitter, relatively high concentrations of OxOrg are predicted for
the non-infected poplar stand (Fig. 8c). In mountain birch stands, the concentration of OxOrg is up to 2-2.5 times higher in
autumnal moth infested stands during the feeding period than in a non-infested birch stand (Fig. 9d). The difference in OxOrg
concentration between moderately and severely infested birch stands is small, due to the combined effects of stress response
(which is a function of the degree of damage) and LAI reduction (Fig. 9d), but towards the conclusion of the feeding period,
the concentration of OxOrg is significantly higher in the less defoliated stand.

### 3.1.3 Formation of new particles

New particles are assumed to be formed from OxOrg and sulfuric acid (Sec. 2.7, Eq. 2). Since the concentration of sulfuric
acid is constrained within the simulations, the concentration pattern of OxOrg is reflected in the seasonal pattern of the
formation rates (Figs. 6c-e, 7f, 8f, 9h,i). Thus, in case of gypsy moth infested oak, the formation rates are much higher (increase
by up to a factor of ~5 ($J_2$), ~7 ($J_3$), ~11 ($J_{10}$)) in stressed stands than in non-infested stands during the period when the plants
are exposed to stress (Figs. 6c-e). The predicted $J_2$ in gypsy moth infested oak stands is comparable to e.g. observations from
Melpitz, Germany ($\sim 9.4 \text{ cm}^{-3} \text{ s}^{-1}$, Paasonen et al., 2010) and San Pietro Capofiume, Italy ($\sim 13 \text{ cm}^{-3} \text{ s}^{-1}$, Paasonen et al.,
2010). Both Melpitz and San Pietro Capofiume are rural sites influenced by anthropogenic pollution (Paasonen et al., 2010).
The modelled $J_2$ in a non-infested oak stand is comparable to observations from Hohenpeissenberg ($\sim 2.3 \text{ cm}^{-3} \text{ s}^{-1}$, Paasonen
et al., 2010) and similar or even higher than typical formation rates measured in the boreal Scots pine forest in Hyytiälä,
Finland ($\sim 1-2 \text{ cm}^{-3} \text{ s}^{-1}$, Paasonen et al., 2010; Kulmala et al., 2012, 2013; Vana et al., 2016) and in the hemiboreal forest
in Järvseljä, Estonia ($\sim 1.09 \pm 1.06 \text{ cm}^{-3} \text{ s}^{-1}$, Vana et al., 2016). By analysing data from Hyytiälä and Järvseljä, Vana et al.
(2016) showed that the values of $J_3$ were in general about 60-80 % of those of $J_2$. Our simulations of a non-infested oak stand





follow this threshold, thus, the predicted $J_3$ in a non-infested oak stand is somewhat higher than observations from Hyytiälä
($\sim 0.6$ cm$^{-3}$ s$^{-1}$, Kulmala et al., 2012, 2013; Nieminen et al., 2014; Vana et al., 2016) and Järvseljä ($\sim 0.8$ cm$^{-3}$ s$^{-1}$, Vana
et al., 2016). $J_3$ in gypsy moth infested oak stands is high, but similar values have occasionally been observed in Hyytiälä (up
to about 10 cm$^{-3}$ s$^{-1}$, Nieminen et al., 2014). Formation rates of 5 nm particles ($J_5 = 1.0 \pm 1.1$ cm$^{-3}$ s$^{-1}$, Yu et al., 2014)
measured in an oak forest in Missouri, USA, are much less than $J_{10}$ in our simulated infested stands, and so are e.g. also
formation rates of 10 nm particles ($J_{10} = 1.2$ cm$^{-3}$ s$^{-1}$, Yli-Juuti et al., 2009) measured in a mixed forest in K-puszta, rural
Hungary. Thus, the predicted formation rates in a non-infested oak stand are comparable, and in the case of gypsy moth infested
oak stands, often much higher than observations from forests with intense new particle formation events.

The formation rates of new particles are always higher in oak powdery mildew infected oak stands than in a non-

infected oak stand (Fig. 7f), though the fungus is not able to perturb the formation rates as strongly (increase by up to a factor
of ~2.3 ($J_2$), ~3.0 ($J_3$), ~5.3 ($J_{10}$)) as herbivory by gypsy moth larvae.

Simulations of poplar stands suggest that particles will be formed at high rates in the range $\sim 3.6 - 11.4$ cm$^{-3}$ s$^{-1}$

($J_2$) and $\sim 2.7 - 10.6$ cm$^{-3}$ s$^{-1}$ ($J_3$) during the late summer when the full leaf state has been attained, and our simulations
suggest that new particles will be formed the fastest in severely rust infected stands (increase by up to a factor of ~3.2 ($J_2$),
~3.9 ($J_3$)).

In our simulations, herbivory by autumnal moth induces the formation rates of new particles in mountain birch stands

by up to a factor of ~2.5 ($J_2$) and ~2.6 ($J_3$). The formation rates of 2 and 3 nm particles are predicted to vary between
0.38 cm$^{-3}$s$^{-1}$ and 2.5 cm$^{-3}$s$^{-1}$ ($J_2$), and 0.31 cm$^{-3}$s$^{-1}$ and 2.5 cm$^{-3}$s$^{-1}$ ($J_3$) in stressed stands, and between 0.32 cm$^{-3}$s$^{-1}$
and 1.1 cm$^{-3}$s$^{-1}$ ($J_2$), and 0.26 cm$^{-3}$s$^{-1}$ and 0.99 cm$^{-3}$s$^{-1}$ ($J_3$) in non-infested stands. The higher end of these values is
comparable to rates observed in Hohenpeissenberg and Hyytiälä (see above). Kyrö et al. (2014) reported that the monthly
averaged formation rate of 3 nm particles during 2005 - 2011, at the SMEAR I station in Värriö, varied throughout the year by
$0.04 - 0.45$ cm$^{-3}$ s$^{-1}$, and by $0.16 - 0.23$ cm$^{-3}$ s$^{-1}$ during the summer months. Analysis of year 2013 and 2014, also in
Värriö, lead to a median formation rate of $0.14 \pm 0.05$ cm$^{-3}$ s$^{-1}$ ($J_3$) (Vana et al., 2016). Thus, the predicted formation rates
in, especially, non-infested mountain birch stands in Lapland are generally within range, but often somewhat higher than
observations from the same location. It should, though, be mentioned that these literature values cannot be used to validate our
simulation results, since Scots pines, and not mountain birches, dominate the SMEAR I site (Kyrö et al., 2014), and the LAI
of mountain birches at the station is significantly less than 2 m$^2$ m$^{-2}$ (Ylivinkka et al., 2020). The modelled formation rates are
not compared to observations from the mountain birch dominated areas in Lapland, since such observations do not, to our
knowledge, exist.

A very recent investigation of long-term field observations (25 years) from the SMEAR I station (Ylivinkka et al.,

2020), where autumnal moth larvae are prominent defoliators of mountain birches, did, however, not find any evidence that
herbivory by autumnal moth would enhance the formation, nor growth, of atmospheric aerosol particles during the summer of
infestation. Instead there was some evidence of elevated total particle concentrations for a few years after the summer with
larval infestation, which was speculated to be caused by delayed defense responses of mountain birches. It is, however, possible
that the total foliage mass of mountain birches in the area is too small, or that the level of infestation was too low during the
investigated time period, in order to cause detectable changes in aerosol variables (Ylivinkka et al., 2020).

The amount of newly formed particles is predicted to be up to about one order of magnitude higher in a gypsy moth

infested oak stand than in a non-infested oak stand, with a 10:00-16:00 median of up to $\sim 1.4 \cdot 10^5$ cm$^{-3}$ in an infested stand
(Fig. 6f). Such a high production of new particles is comparable to observations from e.g. Melpitz (Größ et al., 2018). The
number of produced particles in a non-infested oak stand ($\sim 1.1 \cdot 10^4$ cm$^{-3}$; Fig. 6f) is comparable to e.g. the number of new
particles produced during a typical new particle formation event in Hyytiälä ($\sim 1 - 2 \cdot 10^4$ cm$^{-3}$; Dal Maso et al., 2008;
Nieminen et al., 2014), but significantly higher than observations from a Missouri oak forest, where sub-5 nm particles were
measured to be up to $\sim 2 \cdot 10^4$ cm$^{-3}$, and 5-25 nm particles to $\sim 3000$ cm$^{-3}$, during typical new particle formation events
(Yu et al., 2014). After the period of stress, the number of particles in the growing mode is predicted to range between $\sim 7 \cdot$
$10^3 \, \text{cm}^{-3}$ and $\sim 17 \cdot 10^3 \, \text{cm}^{-3}$ in a non-infested stand, $\sim 6 \cdot 10^3 \, \text{cm}^{-3}$ and $\sim 12 \cdot 10^3 \, \text{cm}^{-3}$ in a 30 % defoliated stand and
between $\sim 3 \cdot 10^{-3} \, \text{cm}^{-3}$ and $\sim 5 \cdot 10^{-3} \, \text{cm}^{-3}$ in a 80 % defoliated oak stand (Fig. 6f). Oak powdery mildew is predicted to
enhance the number of particles in the growing mode by up to a factor of ~4 compared to the corresponding non-infected
stand, resulting in a maximum of $\sim 1.7 \cdot 10^4 \, \text{cm}^{-3}$ in an infected stand, under the used border conditions (Fig 7g). Under the
same environmental conditions, a severely poplar rust infected poplar stand is predicted to produce up to about five times as
many new particles as a non-infected poplar stand, leading to a maximum of about $1.1 \cdot 10^5 \, \text{cm}^{-3}$ in a severely infected stand
(Fig 8h). Finally, it is predicted that herbivory by autumnal moth enhances the amount of produced particles by up to a factor
of ~2.7, with a maximum number of particles in the growing mode of $\sim 3 \cdot 10^4 \, \text{cm}^{-3}$ in an infested birch stand (Fig. 9j). The
predicted amount of particles in a non-infested mountain birch stand is in the same order as observations from Finnish Lapland
(Komppula et al., 2006).

**3.1.4 New particle growth**
New particles are assumed to grow by sulfuric acid and OxOrg (Sec. 2.7, Eq. 6-9), hence the seasonal patterns of formation
rates and OxOrg concentration are reflected in the pattern of the growth rates (Figs. 6g, 7h, 8g, 9k), and therefore also in the
season pattern of the number (Figs. 6f, 7g, 8h, 9j) and size (Figs. 6h, 7g, 8g, 9l) of the growing particle mode. We predict that
the 10:00-16:00 median growth rate in a gypsy moth infested oak stand is at maximum $\sim 5.9 \, \text{nm h}^{-1}$ under the assumed
boundary conditions, whereas the corresponding growth rate in a non-infested oak stand is around $1.6 \, \text{nm h}^{-1}$, when the full
leaf state has been attained (Fig. 6g). For comparison, the growth rate of new particles has been reported to range from 0.5 to
$12 \, \text{nm h}^{-1}$ in Hyytiälä (Dal Maso et al., 2007), with median values of $2.1 \, \text{nm h}^{-1}$ (Vana et al., 2016), $2.5 \, \text{nm h}^{-1}$ (Dal Maso
et al., 2007), and $3.3 \, \text{nm h}^{-1}$ (Paasonen et al., 2010), depending on which years the data covered. Dal Maso et al. (2007)
reported that the growth rate of new small particles in Aspvreten, a rural site in Sweden dominated by deciduous and conifer
forests and some farmlands, ranged between 1 and $11 \, \text{nm h}^{-1}$, with a median value of $3.4 \, \text{nm h}^{-1}$. The growth rate was found
to range from 2.1 to $22.9 \, \text{nm h}^{-1}$, with a median value of $7.25 \, \text{nm h}^{-1}$ during spring in a mixed deciduous forest area close
to Heidelberg in Germany, under influence of anthropogenic pollution (Fiedler et al., 2005). Growth rates from an oak forest
in Missouri, USA, were in the range $1.6 - 11.2 \, \text{nm h}^{-1}$ (Yu et al., 2014). Median values for the growth rate have been reported
to be $4.2 \, \text{nm h}^{-1}$ in Melpitz (Paasonen et al., 2010), $4.6 \, \text{nm h}^{-1}$ in Järvseljä (Vana et al., 2016), $4.8 \, \text{nm h}^{-1}$ in
Hohenpeißenberg (Paasonen et al., 2010) and $9.5 \, \text{nm h}^{-1}$ in San Pietro Capofiume (Paasonen et al., 2010). Thus, we can
conclude that our predicted growth rates are comparable to atmospheric observations from several different rural sites. Growth
rates obtained from areas influenced by anthropogenic pollution are generally higher than our simulated rates, but this is
expected, since our model is constrained by conditions representative for rural sites.
Growth rates are predicted to be lower in an oak powdery mildew infected oak forest, than in a gypsy moth infested
oak forest. The rates are predicted to, at maximum, be $\sim 2.0 \, \text{nm h}^{-1}$ (80 % of leaf area covered by mildew), $\sim 1.6 \, \text{nm h}^{-1}$
(30 % of leaf area covered by mildew) and $\sim 1.2 \, \text{nm h}^{-1}$ (non-infected, in the same environmental conditions as the infected
trees) (Fig. 7h). Thus, the growth rates are similar to the lower end of the observed range.
The growth of small particles in non-infected and rust infected poplar stands are predicted to range between $\sim 2.1$
and $\sim 5.7 \, \text{nm h}^{-1}$, during the late summer when the full leaf state has been attained, with the fastest growth in a heavily rust
infected forest stand (Fig. 8g). This range in growth rates is thus similar to simulation results of herbivory infested oak (see
above; Fig. 6g).
The predicted growth rates are smallest in simulations of non-infested mountain birch stands in Lapland. The 10:00-
16:00 median growth rate is at maximum predicted to be $\sim 4.1 \, \text{nm h}^{-1}$ in an infested stand and varies between $\sim 0.6 \, \text{nm h}^{-1}$
and $\sim 2.0 \, \text{nm h}^{-1}$ in a non-infested stand (Fig. 9k). These values are in line with observations from Värriö (median: $1.6 \pm$



$0.9 \text{ nm h}^{-1}$, Vana et al., 2016; monthly summer mean: $3.7 - 4.4 \text{ nm h}^{-1}$, Kyrö et al., 2014; range: $1 - 10 \text{ nm h}^{-1}$, median:
$2.4 \text{ nm h}^{-1}$, Dal Maso et al., 2007) and from Pallas, Finnish Lapland (median: $2.0 \text{ nm h}^{-1}$, daily range: $0.5 - 11 \text{ nm h}^{-1}$, Dal
Maso et al., 2007; monthly range: $1.9 - 4.6 \text{ nm h}^{-1}$, Asmi et al., 2011).
According to our predictions, new particles will grow up to about 46 nm larger in an oak gypsy moth infested oak
stand compared to a non-infested oak stand within one day (Fig. 6h). Simulation results for the other species/stressors show
that new particles will grow up to about 8 nm more in an oak powdery mildew infected stand (Fig. 7g), $\sim 28$ nm more in a
poplar rust infected poplar stand (Fig. 8g), and $\sim 26$ nm larger in an autumnal moth infested mountain birch stand (Fig. 9l),
within one day, compared to their corresponding non-infected stands. In our simulations, the newly formed particles in non-
infected oak stands are always mainly formed and grown by sulfuric acid (Figs. 6h, 7g), but in modelling of non-infected
poplar, more than half of the formation and growth is due to HOM originating from isoprene (Fig. 8g), while HOM formed
from monoterpenes account for a large fraction of the predicted formation and growth in non-infested birch stands (Fig. 9l).

### 534    3.1.5   R: the isoprene-to-monoterpenes carbon concentration ratio

Previous chamber studies (Kiendler-Scharr et al., 2009, 2012; McFiggans et al., 2019; Heinritzi et al., 2020) have suggested
that isoprene suppresses the formation of new particles from monoterpenes when the isoprene-to-monoterpene carbon
concentration ratio (R) becomes too high. New particle formation has rarely been observed in the field when R>1 (e.g.
Kanawade et al., 2011; Pöschl et al., 2010; Pöhlker et al., 2012; Lee et al., 2016). For example, Yu et al. (2014) observed
formation of sub-5 nm particles during 64 % of the measured days in an oak forest, though R was $15.3 \pm 7.2$ during the
campaign period. However, since the formation of new particles occurs on a regional scale, the authors suggested that the
detected particles could have been formed at lower R and advected to their measurement site. Contrarily, it has earlier been
proposed that oxidation products of isoprene (e.g. IEPOX) promote the growth of existing new particles ($D_p > 3$ nm, e.g.
Surratt et al., 2010; Lin et al., 2013), while Heinritzi et al. (2020) observed the growth of particles above 3.2 nm to be unaffected
by the concentration of isoprene. It is thus likely that the zone of R values, inside which the probability for new particle
formation to occur changes, is influenced by other environmental factors and is therefore location and/or season dependent.
New particle formation has also been observed in the upper troposphere in tropical regions (Andreae et al., 2018;
Williamson et al., 2019) where isoprene dominates the emission spectrum greatly. The hypothesis is that isoprene is vertically
transported via strong convection and new particles are formed from isoprene oxidation products, which is possible due to
lower temperature conditions in the upper troposphere.
The concentration ratio of isoprene-to-monoterpene carbon is very high in non-infected oak and poplar stands and in
oak stands which are no longer exposed to herbivory (Figs. 6b, 7d, 8d), and it is therefore questionable whether particles will
be formed at all in the atmospheric boundary layer from these stands when they are not experiencing stress. Biotic stress greatly
reduces R in all three cases. R is most significantly decreased to a minimum 10:00-16:00 median value of 0.004 in simulations
of gypsy moth infested oak stands (Fig. 6b), but the period with low R values is rather short. For example, R < 1 during only
11 and 4 days, respectively, while R < 22.5 (probably the highest ratio at which new particles formation has been observed in
the field, Yu et al., 2014) during 32 and 21 days, respectively, in our simulations of a severely and moderately European gypsy
moth infested oak stand (Fig. 12e). R is predicted to be close to 1, though never below 1, in simulations of both oak powdery
mildew infected oak stands and rust infected poplar stands. The duration where R is e.g. less than 22.5 is 39 days in a severely
mildew infected oak stand, 31 days in a moderately mildew infected oak stand, and 27 days in a severely rust infected poplar
stand (Fig. 12e). For comparison, R is never predicted to be less than 22.5 in a moderately infected poplar stand (Fig. 8d).
Even if new particles are not formed from oak powdery mildew or poplar rust infected stands in the boundary layer, then both
the potential to form new particles in the upper troposphere (Figs. 7f,g, 8f,h) and the potential to grow already existing particles,
which are formed in nearby stands and horizontally transported to the infected stands (Figs. 7g,h, 8g), are still predicted to be





greater than in our simulations of the correspondingly non-infected stands. R is not relevant in the case of mountain birch,
since this tree species does not emit isoprene constitutively, nor in response to herbivory stress by autumnal moth larvae (Yli-
Pirilä et al., 2016; Rieksta et al., 2020).

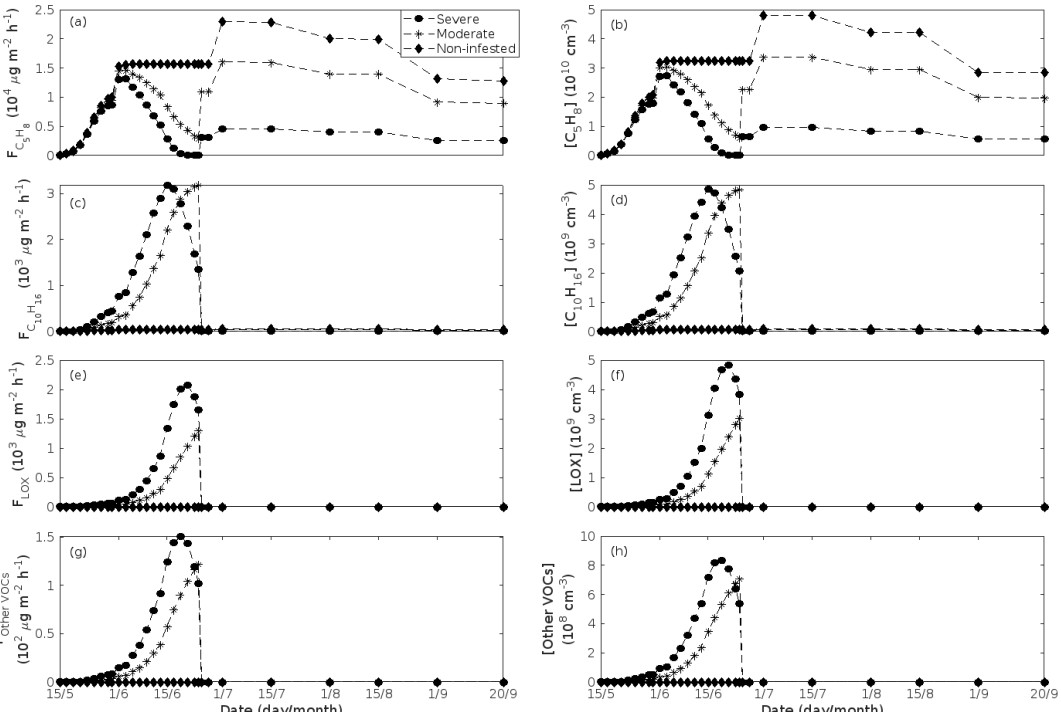


**Figure 5.** A pure oak stand infested with European gypsy moth larvae in comparison to a non-infested pure oak stand. Canopy
emissions of **(a)** isoprene, **(c)** monoterpenes, **(e)** lipoxygenase pathway volatiles (LOX), and **(g)** the sum of other VOCs which
contribute to OxOrg formation (here i.e. methyl salicylate and dimethyl-nonatriene). Atmospheric concentrations of **(b)**
isoprene, **(d)** monoterpenes, **(f)** lipoxygenase pathway volatiles, and **(h)** the sum of other VOCs which contribute to OxOrg
formation. "Moderately" and "severely" refer to 30 % and 80 %, respectively, of the leaf area that has been consumed by the
end of the feeding period.



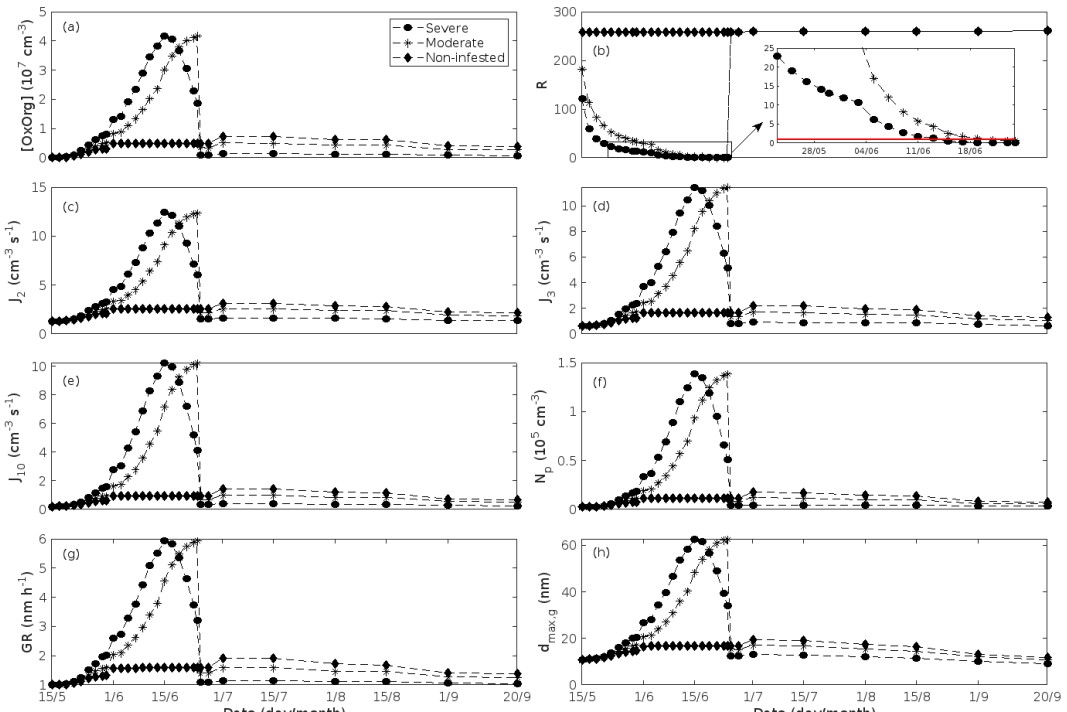

**Figure 6.** A pure oak stand infested with European gypsy moth larvae in comparison to a non-infested pure oak stand. **(a)** atmospheric concentrations of OxOrg. **(b)** the ratios of isoprene-to-monoterpenes carbon concentrations, where the red line indicates R = 1. Formation rates of **(c)** 2, **(d)** 3 and **(e)** 10 nm particles. **(f)** number concentrations of formed particles, **(g)** growth rates of newly formed particles, and **(h)** the daily maxima diameter of the growing particle mode. "Moderately" and "severely" refer to 30 % and 80 %, respectively, of the leaf area that has been consumed by the end of the feeding period.






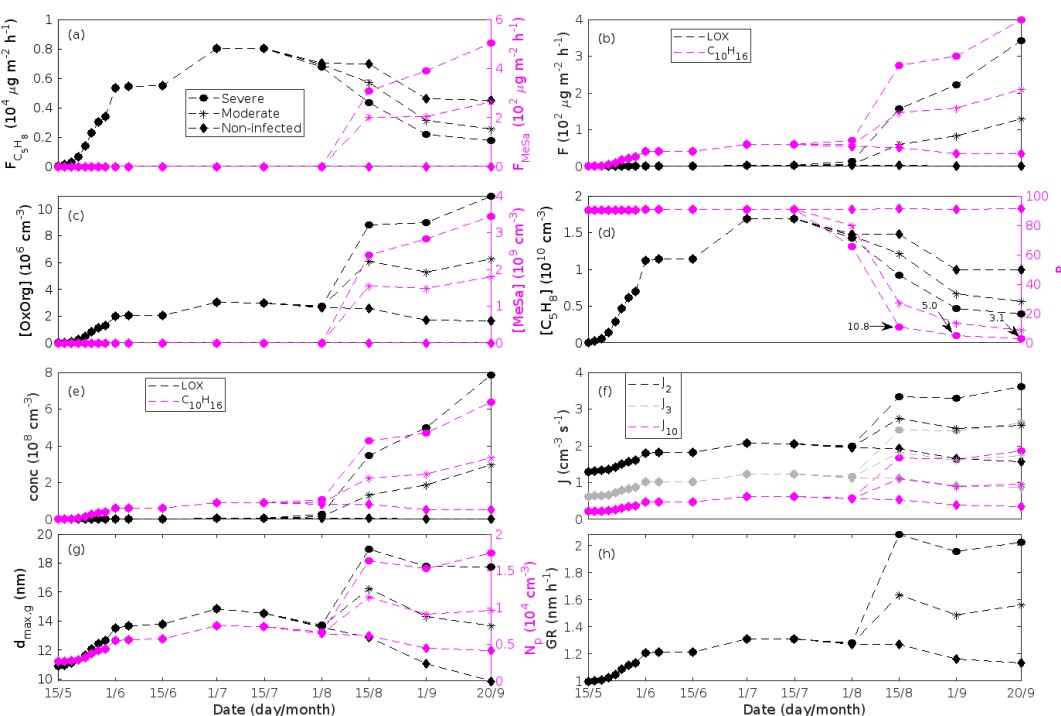


**Figure 7.** A pure oak stand infected by oak powdery mildew in comparison to a non-infected pure oak stand. Canopy emissions
of **(a, left axis)** isoprene, **(a, right axis)** methyl salicylate, **(b)** lipoxygenase pathway volatiles and monoterpenes. Atmospheric
concentrations of **(c, left axis)** OxOrg, **(c, right axis)** methyl salicylate, **(d, left axis)** isoprene, and **(e)** lipoxygenase pathway
volatiles and monoterpenes. **(d, right axis)** the ratios of isoprene-to-monoterpene carbon concentrations. **(f)** formation rates of
2, 3 and 10 nm particles. **(g, left axis)** daily maxima diameter of the growing particle mode, and **(g, right axis)** number
concentrations of formed particles. **(h)** growth rates of newly formed particles. "Moderately" and "severely" refer to 30 % and
80 %, respectively, of the leaf area that has been infected by fungi by the onset of senescence.






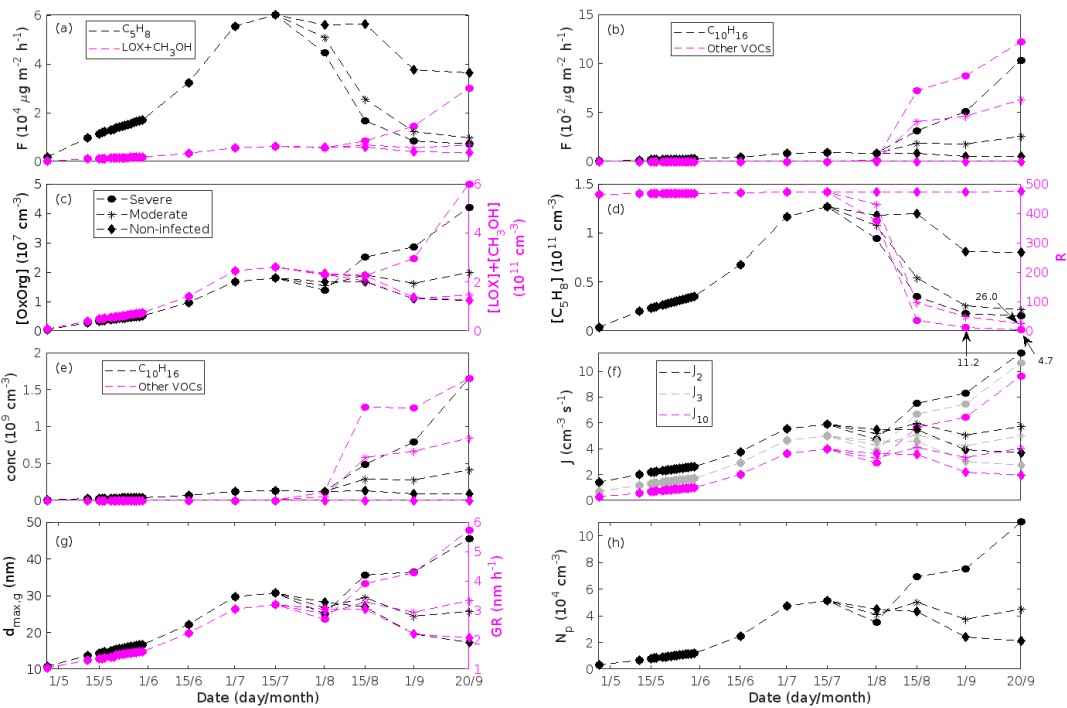


**Figure 8.** A pure poplar stand infected by rust fungi in comparison to a non-infected pure poplar stand. Canopy emissions of
**(a)** isoprene and the sum of lipoxygenase pathway volatiles and methanol, **(b)** monoterpenes and the sum of other VOCs which
contribute to OxOrg formation (here i.e. methyl salicylate, dimethyl-nonatriene, α-Eudesmol and sesquiterpenes). Atmospheric
concentrations of **(c, left axis)** OxOrg, **(c, right axis)** the sum of lipoxygenase pathway volatiles and methanol, **(d, left axis)**
isoprene, and **(e)** monoterpenes and the sum of other VOCs which contribute to OxOrg formation. **(d, right axis)** the ratios of
isoprene-to-monoterpene carbon concentrations. **(f)** formation rates of 2, 3 and 10 nm particles. **(g, left axis)** daily maxima
diameter of the growing particle mode, and **(g, right axis)** growth rates of newly formed particles. **(h)** number concentrations
of formed particles. "Moderately" and "severely" refer to 30 % and 80 %, respectively, of the leaf area that has been infected
by fungi by the onset of senescence.





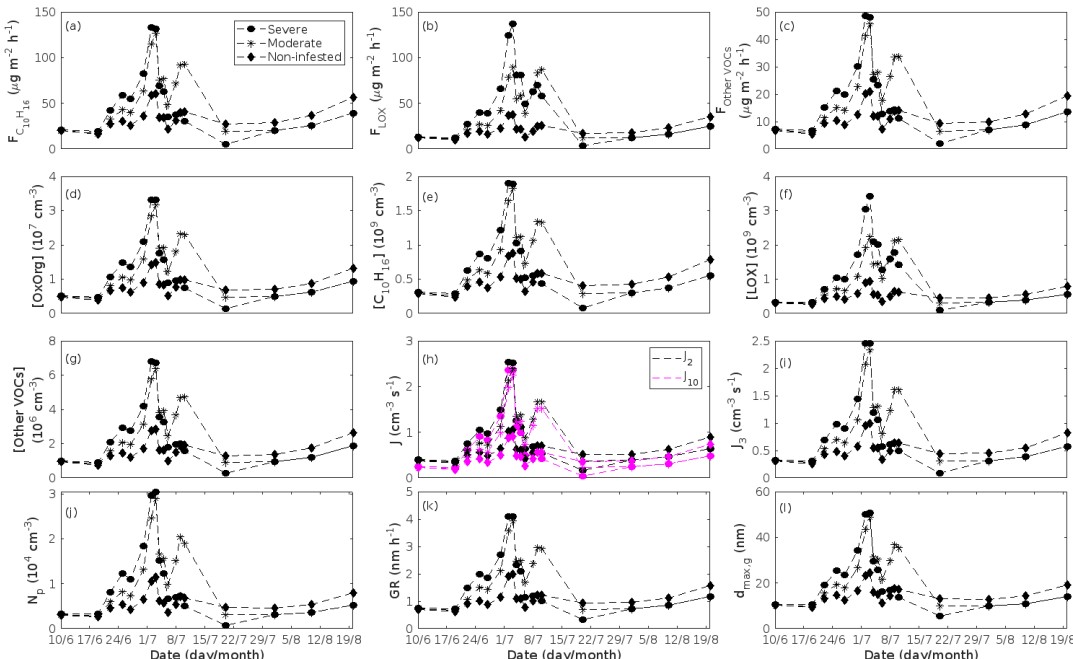

**Figure 9.** A pure mountain birch stand infested with autumnal moth larvae in comparison to a non-infested pure mountain
birch stand. Canopy emissions of **(a)** monoterpenes, **(b)** lipoxygenase pathway volatiles, and **(c)** the sum of other VOCs which
contribute to OxOrg formation (here i.e. dimethyl-nonatriene and sesquiterpenes). Atmospheric concentrations of **(d)** OxOrg,
**(e)** monoterpenes, **(f)** lipoxygenase pathway volatiles, and **(g)** the sum of other VOCs which contribute to OxOrg formation.
Formation rates of **(h)** 2, 10 and **(i)** 3 nm particles. **(j)** number concentrations of formed particles. **(k)** growth rates of newly
formed particles. **(l)** daily maxima diameter of the growing particle mode. "Moderately" and "severely" refer to 30 % and 80
%, respectively, of the leaf area that has been consumed by the end of the feeding period.

**3.2 Estimating the reliability of our results**
Since aerosol processes are very sensitive to changes in environmental conditions - conditions which can vary greatly, both
interannually, but also from day to day, we investigated the influence of a wide range of realistic and relevant environmental
conditions (Table C1 in Appendix C) on our model predictions (Figs. 10-11, C1-2 in Appendix C). Nine different sensitivity
tests (ST1-9) were conducted for all plant species and infections, where only one parameter was changed at a time (Table C1).
For these simulations, the default values listed in Table 2 were used, while the default maximum daily temperature at
Hohenpeissenberg and SMEAR I were assigned to 25 °C and 20 °C, respectively, and the default LAI for oak/poplar and birch
was assumed to be 5 $m^2$ $m^{-2}$ and 2 $m^2$ $m^{-2}$, respectively. All aerosol parameters (formation and growth rates, diameter, number
of particles) show a similar response to changes in the considered environmental parameters, thus only the impact on the
number of newly formed particles (Figs. 10-11) and the rate at which new small particles grow (Figs. C1-2) is displayed.
As is also observed in nature, certain conditions suppress or prevent the formation of new particles, such as for
example a high condensation sink (Fig. 11d,i,n,s; e.g. Hyvönen et al., 2005; Nieminen et al., 2015; Vana et al., 2016) and low
sulfuric acid concentration (Fig. 11c,h,m,r; e.g. Boy et al., 2005; Nieminen et al., 2014), making the atmospheric relevance of
the forest stands minor. Since we have assumed realistic conditions, but at the same time conditions which do not prevent the
formation of new particles, in our simulations, the number of predicted days with occurring new particle formation is the





theoretical maximum for clean environments, which our aerosol theory is based on (Sec. 2.7). Though the absolute number of
predicted new particles depend highly on the assumed environmental conditions (Figs. 10-11), the relative difference between
non-infected and stressed stands of the same tree species is not impacted: e.g. the number of new particles is always
significantly higher in gypsy moth infested, and oak powdery mildew infected, oak stands, than in non-infected oak stands,
when the environmental conditions are assumed to be the same in all stands (Figs. 10a-h, 11a-j). Likewise, more particles are
always formed in moderately, than severely, moth infested oak and birch stands, since the decrease in LAI is stronger than the
increase in the stress-induced emission response per unit leaf area (Figs. 10a-d,m-p, 11a-e,p-t). This is emphasised in very
severely infested mountain birch stands (e.g. 80 % defoliation), where the number of produced particles is always less than in
its corresponding non-infested stand (Figs. 10m-p, 11p-t).

Sensitivity tests were also carried out in order to assess whether the simplifications made in the model are valid: (1)

As mentioned earlier (Sec. 2.4), we did not incorporate a full canopy environment in the model - an approach which has also
been taken by other investigators (e.g. Simpson et al., 1999, 2012; Bergström et al., 2014). In ST2 (Table C1, Figs. 10b,f,j,n,
C1b,f,j,n) changes in light conditions exclusively impact the predicted emissions of VOCs. From Figs. 10b,f,j,n, C1b,f,j,n it is
clear that even assuming extremely different light environments would not change our conclusions about the atmospheric
importance of biotic plant stresses, since our results show that stressed forest with a maximum light availability down to 200
$\mu$mol m$^{-2}$ s$^{-1}$ would still produce more new particles than its correspondingly non-infected stand at theoretically clear sky
conditions (Fig. 10b,f,j,n). A highly autumnal moth stressed mountain birch stand (80 % defoliation) would possibly produce
slightly more particles than a non-infested stand, if a full canopy environment would be considered. For example, the number
of produced particles is slightly higher in a birch stand experiencing a stress level of 80 % under 1000 $\mu$mol m$^{-2}$ s$^{-1}$ than a non-
infested stand under 400 $\mu$mol m$^{-2}$ s$^{-1}$ (Fig. 10n). However, the LAI of mountain birch stands is usually rather low (Heiskanen,
2006), making the difference in light environment between a non-infested and a highly defoliated stand small. Since mildew
and rust do not decrease the leaf area of their host, a different treatment of the light environment would not influence the
relative atmospheric importance of fungally infected oak and poplar vs their correspondingly non-infected stands (Fig. 10f,j).

(2) In ST3 (Table C1, Figs. 10c,g,k,o, C1c,g,k,o) and in our seasonal simulations (Figs. 5-9), the change of

temperature only impacts the emission rates of VOCs. In reality, the vapour pressures of oxidised compounds increase non-
linearly with an increase in temperature (e.g Bilde et al., 2015), and less HOM, and other oxidised organic compounds, will
therefore condense at higher temperatures, whereby the formation and subsequent growth of particles will decrease
(Stolzenburg et al., 2018; Simon et al., 2020). Gas phase chemistry, including the formation of HOM, is also in reality
temperature dependent (e.g. Quéléver et al., 2019). These effects have not been included in the model. Since the range of daily
maximum temperatures throughout the growing season is assumed to be rather narrow (Fig. 4), this effect does not greatly
impact our results (Sec. 3.1), but it means that the number of particles produced at high temperatures (Fig. 10c,g,k,o), and the
growth rate at which they are produced (Fig. C1c,g,k,o), are overestimated for both non-infected and stressed forests.

(3) The concentrations of ozone and OH were unaltered between simulations of non-infected forests and forests under

varying degrees of infection (Sec. 2.6), though in reality, the atmospheric oxidation capacity is controlled by changes in the
concentration of atmospheric trace gases, including VOCs. The total emission of VOCs from oak and poplar stands is greatly
dominated by isoprene, but the emission of isoprene decreases as a function of biotic stress severity (Figs. 5a, 7a, 8a). In
contrast, the emission of LOX, methyl salicylate, methanol, monoterpenes and sesquiterpenes increases as the level of stress
increases (Figs. 5c,e,g, 7a,b, 8a,b). The oxidation of isoprene, LOX, methyl salicylate and methanol is primarily driven by
reactions with OH, and also monoterpenes react with OH, which all leads to reductions in the concentration of OH (Table 3),
though e.g. ozonolysis of monoterpenes also produce OH, which thus counters part of the reduction. When considering the
reaction rates and emission rates of the considered VOCs in simulations of oak and poplar stands, the concentration of OH is
mainly controlled by changes in the emission of isoprene. Thus, we expect that the concentration of OH will increase as the
degree of stress increases, but even a strong shift in the concentration of OH, will not change the conclusion about the relative





atmospheric importance of stressed vs stress-free oak and poplar forests (Figs. 11b,g,l,q, C2b,g,l,q). The absolute number of
predicted new particles in herbivory stressed oak stands will, however, be predicted to be smaller at higher levels of OH (Fig.
11b), because the oxidation of monoterpenes is then more strongly controlled by OH, which leads to a smaller production of
HOM, as monoterpenes typically form HOM at a considerably lower yield from reactions with OH than ozone (Appendix B).
A similar shift in the oxidation of monoterpenes is happening in case of oak powdery mildew infected oak, but the effect is
counted by an increase in the formation of oxidised organic compounds from oxidation of methyl salicylate at high levels of
OH, leading totally to higher predicted particle number concentrations (Fig. 11g). Considering the emissions from biotically
stressed and non-infested mountain birch, we estimate that the concentration of OH should stay largely the same, or potentially
decrease slightly at higher levels of infestation, which will enhance the oxidation of monoterpenes by ozone, which will lead
to a larger production of HOM and thereby a slightly higher predicted number of new particles (Fig. 11q). In the atmosphere,
the production of sulfuric acid is limited by the availability of OH, and it is therefore possible that the effects of changes in the
concentration of OH (Fig. 11b,q) and sulfuric acid (Fig. 11c,r), in herbivory stressed stands, on the absolute number of
predicted new particles, will cancel out or even lead to a stronger particle production than predicted. In case of oak powdery
mildew infected oak, the two effects will enhance each other and result in an even higher number of predicted particles. In
clean, low $NO_X$ environments, which we aimed to simulate, the concentration of ozone is largely unaffected by the ambient
concentration of isoprene (e.g. Jenkin et al., 2015). However, isoprene forms ozone progressively with an increased availability
of $NO_X$ (e.g. Jenkin and Clemitshaw, 2000). Higher ozone levels support enhanced formation of HOM, and thus aerosol
processes, but the production of HOM is also known to decrease as a function of increased $NO_X$ concentration (e.g. Ehn et al.,
2014), whereby the formation and growth of new particles becomes suppressed (e.g. Yan et al., 2020; Pullinen et al., 2020).

(4) As mentioned earlier (Sec. 2.6), many HOM yields have not been investigated for the exact compounds which are

emitted from the tree species, which are the focus of this study. From Fig. 11e,j,o,t it is obvious, that even if all the OxOrg
yields used for simulations of only biotically stressed forests were to be decreased significantly - in case of moderately
herbivory infested oak (30 % of leaf area defoliated) by down to about 95 % - biotically stressed oak, poplar and mountain
birch forests would still, in most cases, produce more particles than non-infected forests of the same tree species. The yields,
at which HOM are formed, have been treated as fixed values (Appendix B) in the seasonal simulations (Sec. 3.1), but the yields
actually depend on several factors, such as e.g. the concentration of $NO_X$ (point 3 above; Ehn et al., 2014), temperature (point
3 above; Quéléver et al., 2019; Simon et al., 2020) and the ambient blend of VOCs (Sec. 3.1.5; McFiggans et al., 2019). Exactly
how the formation and growth of new particles depend on the VOC blend is still uncertain, but it has recently been
demonstrated that a linear addition of the yields from the individual yields of components in the VOC mixture will result in an
overestimation of both the number and size of particles (McFiggans et al., 2019). As we have followed a similar procedure,
this effect might cause our predicted aerosol processes (Sec. 3.1) to be overestimated, but since the ratio of isoprene-to-
monoterpenes carbon concentration is much higher in non-infected oak and poplar stands than in the correspondingly stressed
stands (Figs. 6b, 7d, 8d), the overestimation is expected to be more pronounced in the non-infected stands (McFiggans et al.,
2019). The difference in the atmospheric importance of non-infected and biotically stressed oak and poplar stands thereby
widens (Fig. 11e,j,o).

It is well known that the potential for foliage to emit VOCs depends on the age of the foliage: emerging and growing

foliage usually emits isoprene at reduced rates (e.g. Guenther et al., 1991, 2012; Goldstein et al., 1998; Petron et al., 2001) and
monoterpenes at enhanced rates (e.g. Guenther et al., 1991, 2012; Aalto et al., 2014; Taipale et al., 2020) compared to that of
its corresponding mature foliage. Old leaves do usually additionally emit isoprene at decreased rates (Monson et al., 1994;
Schnitzler et al., 1997; Sun et al., 2012). These effects were not considered in our simulations (Sec. 3.1), since the effect of
leaf age on biotic plant stress emissions is unexplored. Considering a similar treatment of the impact of leaf maturity on the
emissions of VOCs as Gunther et al. (2012) (see Appendix A) would only influence the predicted number and size of particles
in herbivory stressed and non-infested oak forests insignificantly (Fig. A2). However, it would decrease the ratio of isoprene-





to-monoterpenes carbon so significantly in gypsy moth infested oak stands, that the possible suppression of aerosol processes
by isoprene would disappear during most, or even the entire, duration of stress (Fig. A2b). Applying a similar leaf age effect
as described in Gunther et al. (2012) on simulations of fungally infected oak and poplar forests would not decrease the ratio of
isoprene-to-monoterpenes carbon sufficiently in order to avoid the possible suppression effect of isoprene, since Guenther et
al. (2012) only assume a reduction of 10 % on the emissions of isoprene from old leaves. We investigated that the emission of
isoprene from mildew infected oak would need to decrease by ~68-96 % (severity of stress ranging from 80 to 9 %), in
comparison to simulations where the leaf age effect is not considered, in order to reach R≤1, whereas the emission of isoprene
from non-infected oak would need to decrease by ~99 % (Fig. A3a). In comparison, the emission of isoprene from a non-
infected poplar stand would need to decline by 99.8 %, and from a rust infected poplar stand by at least 79 %, in order to attain
R≤1 (Fig. A4a). In order to reach R≤22.5, the upper limit at which new particle formation has been observed in the atmosphere
(Yu et al., 2014), the emission of isoprene from a non-infected poplar stand would need to decrease by ~95 %, whereas heavily
rust infected poplar forest would already be below this limit without considering an age dependent reduction of the emission
potential (Fig A4a). Simulations were not done for mountain birch forest stands, since no emissions are suppressed upon
herbivory stress of mountain birch (Yli-Pirilä et al., 2016) and since Yli-Pirilä et al. (2016) did not provide age information on
the leaves they measured.

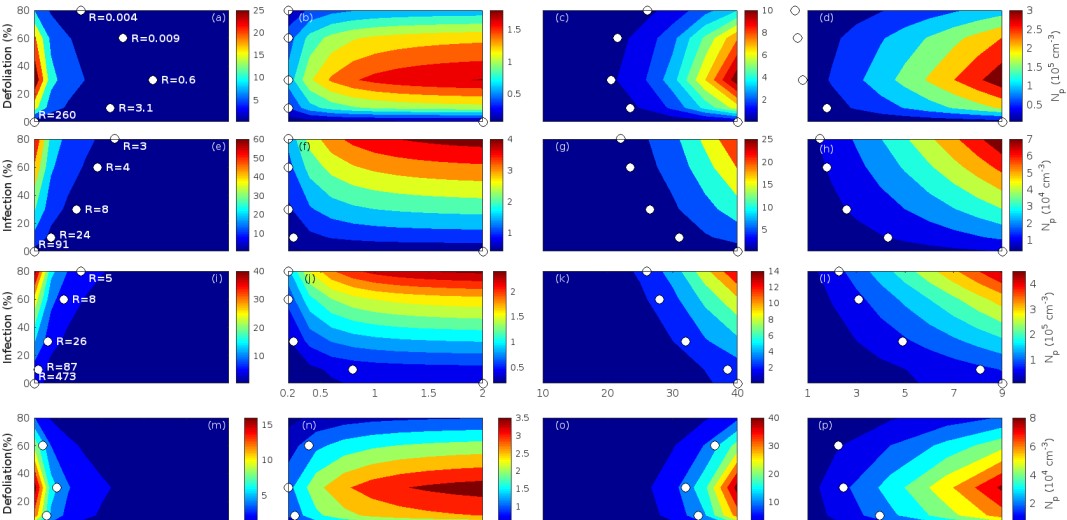

**Figure 10.** Impact of changed boundary conditions on the number concentrations of newly formed particles in non-infected
and biotically stressed forest stands. The number concentration of newly formed particles is expressed as a function of changes
in the boundary layer height **(a, e, i, m)**, light **(b, f, j, n)**, temperature **(c, g, k, o)** and leaf area index **(d, h, l, p)** for non-infected
and infected oak **(a-d**, gypsy moth, **e-h**, powdery mildew), poplar **(i-l)** and birch **(m-p)** stands. Light **(b, f, j, n)** and temperature
**(c, g, k, o)** are given as the daily maxima, but in the simulations the parameters follow a daily cycle. The displayed LAI **(d, h,**
**l, p)** is that of a non-infected stand, hence e.g. at LAI = 7 $m^2$ $m^{-2}$, the simulation for a larval infestation level of 80 % has been
conducted with LAI = 1.4 $m^2$ $m^{-2}$, which is 20 % of the non-infected stand LAI value. Optimal conditions (i.e. leading to
highest number concentrations) for non-infected stands are indicated with white markers at an infection level of 0 %. White
markers located at various infection levels mark the conditions at which an identical or slightly higher number concentration,
as produced by a non-infected forest stand at optimal conditions, is reached. No markers are used for 80 % defoliated mountain
birch **(m-p)**, since the corresponding number concentrations are always lower than in a non-infested birch stand at optimal





conditions. Be aware that white markers in **d** and **p** are not located at the LAI of a non-infected stand, but instead at the values
used for the simulations. R values **(a, e, i)** indicate the ratio of isoprene carbon / monoterpene carbon at the locations of the
write markers. Be aware that the x-axes are different for simulations in Hohenpeißenberg **(a-l)** and SMEAR I conditions **(m-**
**p)** except in the case of changing boundary layer height **(a, e, i, m)**.

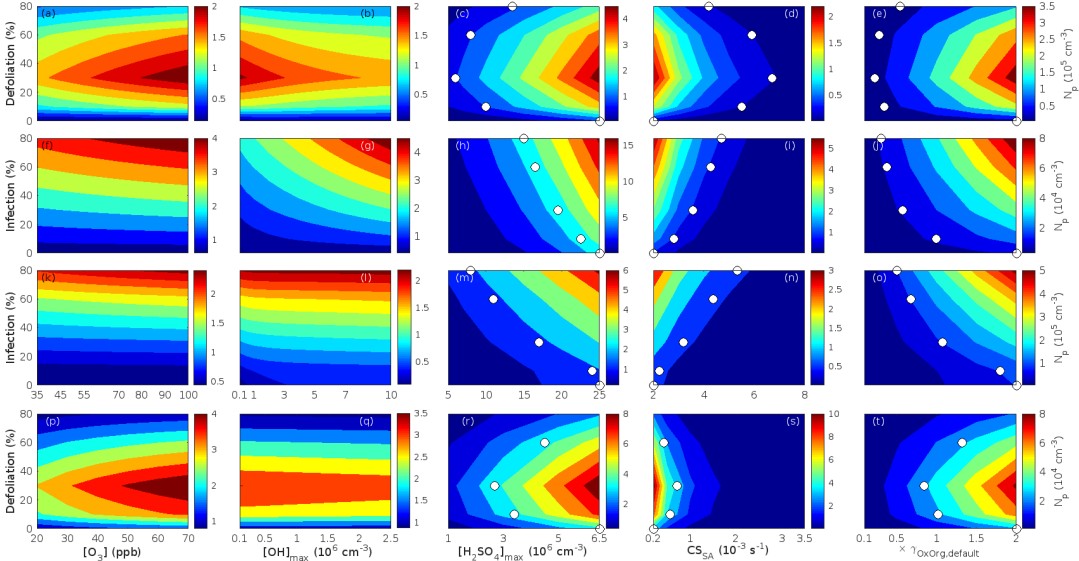


**Figure 11.** Impact of changed boundary conditions on the number concentrations of newly formed particles in non-infected
and biotically stressed forest stands. The number concentration of newly formed particles is expressed as a function of changes
in the concentration of ozone **(a, f, k, p)**, OH **(b, g, l, q)** and sulfuric acid **(c, h, m, r)**, the condensation sink **(d, i, n, s)** and
OxOrg yields **(e, j, o, t)** for non-infected and infected oak **(a-e**, gypsy moth, **f-j**, powdery mildew), poplar **(k-o)** and birch **(p-**
**t)** stands. The concentrations of OH **(b, g, l, q)** and sulfuric acid **(c, h, m, r)** are given as the daily maxima, but in the simulations
the parameters follow a daily cycle. The displayed condensation sink (CS$_{SA}$**: d, i, n, s**) is that of sulfuric acid, while the
condensation sink of OxOrg, in the respective simulations, is half of CS$_{SA}$. White markers are used in a similar way as in Fig.
10. Be aware that the x-axes are different for simulations in Hohenpeißenberg **(a-o)** and SMEAR I conditions **(p-t)** except in
the case of changing Oxorg yields **(e, j, o, t)**.

### 3.3 Implications and remaining issues to be explored

Our simulation results (Figs. 5-9) illustrate that biotic plant stresses are capable of substantially perturbing both the number
and size of atmospheric aerosol particles throughout a significant fraction of the year (summarised in Fig. 12). Considering
that we calculated *daily* new particle growth, our results point to the direction that induced plant emissions will subsequently
lead to more efficient CCN production in the atmosphere (Fig. 12), which will moreover affect cloud properties, such as cloud
albedo and lifetime (Twomey, 1977; Albrecht, 1989; Gryspeerdt et al., 2014; Rosenfeld et al., 2014). The amplitude of the
enhancement, however, depends strongly on the specific stressor and tree species which is attacked.
Naturally, both the duration of stress (Fig. 12e) and the predicted number (Fig. 12d)) and size (Fig. 12c) of new
particles depend highly on our assumptions about e.g. when the fungi start to attack their host, how fast the fungi spread,
whether the larval eggs hatch simultaneously with budburst, how fast larval development occurs, and when senescence onsets
- all which depend strongly on environmental conditions. It is furthermore probable that emissions are also induced from





fungally infected leaves during senescence, which was not simulated here. The duration of stress can, thus, be significantly
longer than what is summarised in Fig. 12e, whereby also the post-defoliation period, in case of herbivory infestations, will be
shorter, and the atmospheric importance of the stresses stronger.
Vegetation is often subject to more than one type of stress simultaneously, which generally enhances the already
induced emission response due to biotic plant stress (e.g. Blande et al., 2007; Vapaavuori et al., 2009; Holopainen and
Gershenzon, 2010; Kivimäenpää et al., 2016; Ghimire et al., 2017). It has, for example, recently been shown that warming
significantly amplifies the emission response due to autumnal moth feeding (Li et al., 2019). Trees which have been subject
to herbivory also have a tendency to more often be severely attacked by fungi later during the summer, than trees which have
not experienced defoliation (e.g. Marçais and Desprez-Loustau, 2014).
Finally, we have shown that it could be more important to account for biotic plant stresses in models than significant
variations in those environmental parameters which predictions of VOC emissions are currently controlled by, e.g. light
conditions (Fig. 10b,f,j,n), temperature (Fig. 10c,g,k,o) and LAI (Fig. 10d,h,l,p). Considering changes in the emissions of
VOCs caused by stress also seems to be more crucial than accounting for large changes in the concentrations of $O_3$ (Fig.
11a,f,k,p) and OH (Fig. 11b,g,l,q).
Together with the facts that insect outbreaks and fungal diseases generally are expected to increase in both frequency
and severity in the future (Cannon, 1998; Bale et al., 2002; Harrington et al., 2007; Pautasso et al., 2012; Boyd et al., 2013),
our findings underline the urgency of accounting for biotic plant stress emissions in numerical models.

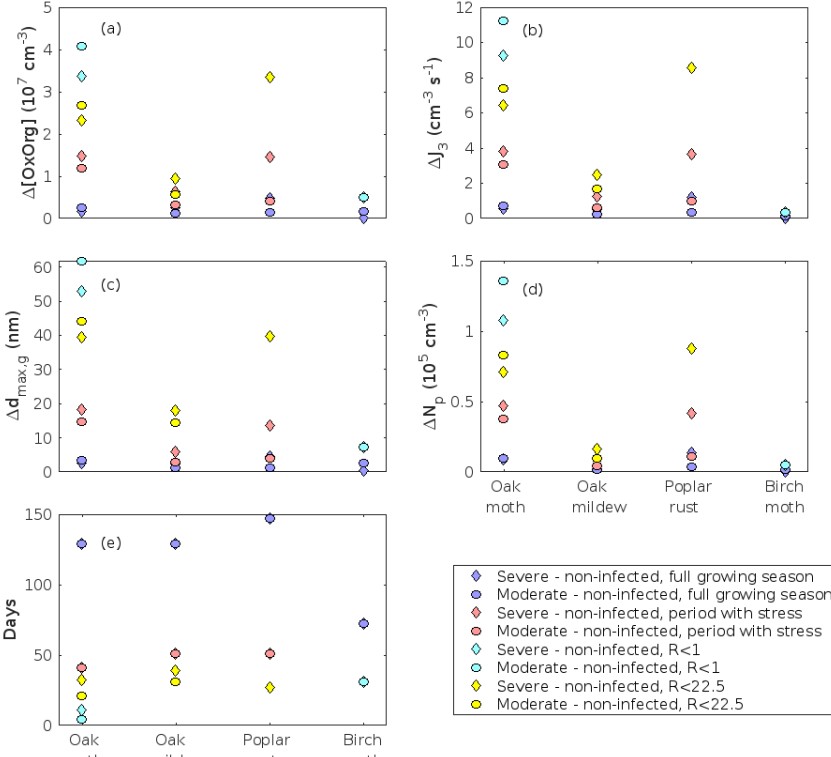


**Figure 12.** Differences in atmospheric response from various non-infected and biotically stressed plant species. **(a)**
atmospheric concentrations of OxOrg, **(b)** formation rates of 3 nm particles, **(c)** daily maximum diameter of the growing




particle mode, **(d)** number concentration of formed particles, and **(e)** amount of days considered. **a-d** are provided as the
differences between the averaged parameter in a stressed and stress-free forest stand of the same plant species type. Differences
and averages are considered based on the complete growing season, the period with stress, when the ratio of isoprene-to-
monoterpenes carbon concentration is less than 1 or less than 22.5. Be aware that R is always zero in simulations of birch. In
cases where R does not reach less than 1 or less than 22.5 in the atmosphere surrounding a non-infected forest stand, but it
does in the case of the corresponding stressed stand, it is assumed that the atmospheric parameter in the non-infected stand is
zero and hence the difference is given as the value of the stressed stand. The concentration differences of OxOrg, formation
rates and number concentrations are calculated based on an average, for the period of interest, of the median values during
10:00-16:00 local time. "Severe" and "moderate" refer to that 80 % or 30 % of the total leaf area has been consumed or infected
by the end of the feeding/infection period, respectively. Southern Germany has been used as border conditions for simulations
of oak and poplar, while SMEAR I, Finnish Lapland, has been used for modelling of birch.

## 4 Conclusions

We constructed a conceptual model to simulate new particle formation and growth in various broadleaved forest stands, in
clean low $NO_X$ environments, under biotically stressed and stress-free conditions, throughout a full growing season.
Unsurprisingly, we found that the predicted atmospheric importance of biotic plant stress highly depends on the specific
individual stressor and tree species which is attacked. Thus, the amount of newly formed particles was predicted to be up to
about one order of magnitude higher in a gypsy moth infested oak stand than in a non-infested oak stand. In comparison, the
number of new particles was simulated to be up to about a factor of 3, 4 and 5 higher in autumnal moth, oak powdery mildew
and poplar rust infected mountain birch, pedunculate oak and balsam poplar stands, respectively. We, furthermore, predicted
that the new particles will grow up to about 46, 28, 26 and 8 nm larger in an oak gypsy moth, poplar rust, autumnal moth and
oak powdery mildew infected stand, respectively, compared to their corresponding non-infected stands within one day. To our
knowledge, this study is the first to investigate the atmospheric impact of biotic plant stresses throughout a full growing season.
Our modelling results generally indicate that all the investigated plant stresses are capable of substantially perturbing
both the number and size of atmospheric aerosol particles, and it is thus likely that the induced emissions will subsequently
lead to more efficient CCN production in the atmosphere. We also showed that it can be more important to account for biotic
plant stresses in models than significant variations in e.g. LAI, and temperature and light conditions, which are currently the
main parameters controlling predictions of VOC emissions. Since insect outbreaks and fungal diseases are generally expected
to increase in both frequency and severity in the future, our findings underline the urgency of accounting for biotic plant stress
emissions in numerical models.

*Data availability*. The model code is available upon reasonable request by contacting ditte.taipale[at]helsinki.fi. SMEAR I
mountain birch leafing data can be obtained by contacting vesa.haataja[at]helsinki.fi. All other data used to constrain the model
is publicly available following the provided references.

*Author contributions*. Idea and concept by ÜN. ÜN standardised the published emission rates. VM, MK and ME developed
the theory for the aerosol module. DT developed the model code, conducted the simulations, and wrote the manuscript, with
inputs from all authors. All authors discussed the results, and commented and edited the manuscript.

*Competing interests*. The authors declare that they have no conflict of interest.





Acknowledgements. This work was supported by the European Regional Development Fund (Centre of Excellence
EcolChange), the European Research Council (advanced grant 322603, SIP-VOL+), the Academy of Finland Center of
Excellence project (no. 307331), and the Academy of Finland Flagship funding (grant no. 337549). DT was also supported by
the Academy of Finland (no. 307957). DT thanks Pekka Kaitaniemi for valuable discussions related to autumnal moth
simulation set-up and Juho Aalto, Yifan Jiang and Tero Klemola for use of photos.

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

**Appendix A: Leaf age effect**
**Simulations of pedunculate oak infested with European gypsy moth larvae**
It seems likely that Copolovici et al. (2017) conducted their measurements on leaves that emit isoprene at peak rate, since their
reported emission rate of isoprene from non-infested leaves is comparable to the rates reported from mature leaves in previous
studies (e.g. Smiatek and Steinbrecher, 2006; Perez-Rial et al., 2009; van Meeningen et al., 2016). The impact of leaf age was
tested during the period with stress utilising the moderations shown in Table A1. The first period covers the number of days
between budbreak and the induction of isoprene emission, while the second period ends when initiation of peak isoprene
emission rates has been reached. The duration of these two periods were calculated using Eq. 18a and Eq. 19 in Guenther et
al. (2006) and our assumptions about the ambient temperature conditions (Fig. 4b). Since isoprene does not show an induced
response in emission upon gypsy moth herbivory, the emission rate of isoprene was reduced in simulations of both non-infested
and infested oak forest. The applied factors used for reductions are from Guenther et al. (2012). The emission rate of
monoterpenes was increased in the beginning of the growing season for simulations of a non-infested oak stand utilising the
coefficients from Guenther et al. (2012). Stress-induced emissions were not altered, since we do not know the effect of leaf
age on these types of emissions and Guenther et al. (2012) e.g. also assumed the same. The results are shown in Fig. A1-2.

**Table A1.** Moderations made in order to consider the effect of leaf age. $\varepsilon_{isoprene}$ and $\varepsilon_{monoterpenes}$ are the emission factors of
isoprene and monoterpenes, respectively, used in the simulations when the leaf age effect has been considered, while $\varepsilon_{iso,mature}$
and $\varepsilon_{mono,mature}$ are the emission factors of isoprene and monoterpenes, respectively, used in the default simulations (i.e.
resulting in Fig. 5-6). The moderations have been applied to the simulations of either only non-infested oak or also stressed
oak as indicated under "simulations".

| Period (day/month) | $\varepsilon_{isoprene}$ | Simulations | $\varepsilon_{monoterpenes}$ | Simulations |
|---|---|---|---|---|
| 15-26/5 | $0.05 \times \varepsilon_{iso, mature}$ | Non-infested, moderate stress, severe stress | $2 \times \varepsilon_{mono, mature}$ | Non-infested |
| 27/5-11/6 | $0.6 \times \varepsilon_{iso, mature}$ | Non-infested, moderate stress, severe stress | $1.8 \times \varepsilon_{mono, mature}$ | Non-infested |
| 12/6- | $\varepsilon_{iso, mature}$ | Non-infested, moderate stress, severe stress | $\varepsilon_{mono, mature}$ | Non-infested, moderate stress, severe stress |




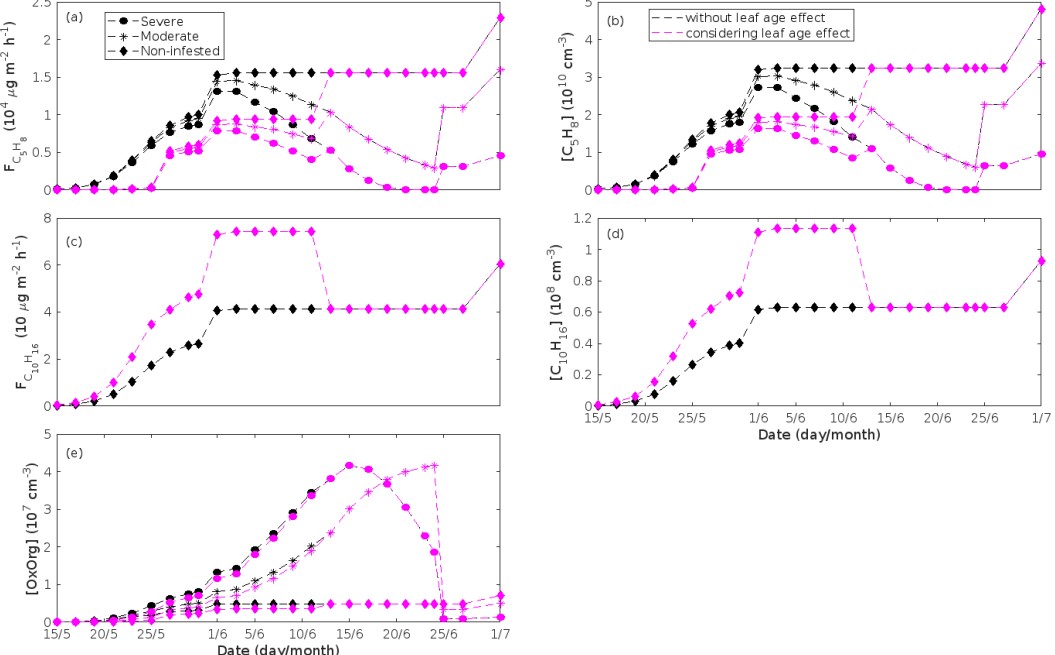

**Figure A1.** An oak stand infested with European gypsy moth larvae in comparison to a non-infested oak stand simulated with and without considering the impact of leaf age on the rates of emissions. Canopy emissions of **(a)** isoprene and **(c)** monoterpenes, atmospheric concentrations of **(b)** isoprene, **(d)** monoterpenes and **(e)** OxOrg. "Moderately" and "severely" refer to 30 % and 80 %, respectively, of the leaf area that has been consumed by the end of the feeding period. Black makers are for simulations where the effect of leaf age was not considered, while magenta markers are for simulations where the effect of leaf age was considered. Simulation results (independently of whether the effect of leaf age was considered or not) for "severe" is always indicated by circles, for "moderate" by asterisks, and for "non-infested" by diamonds.

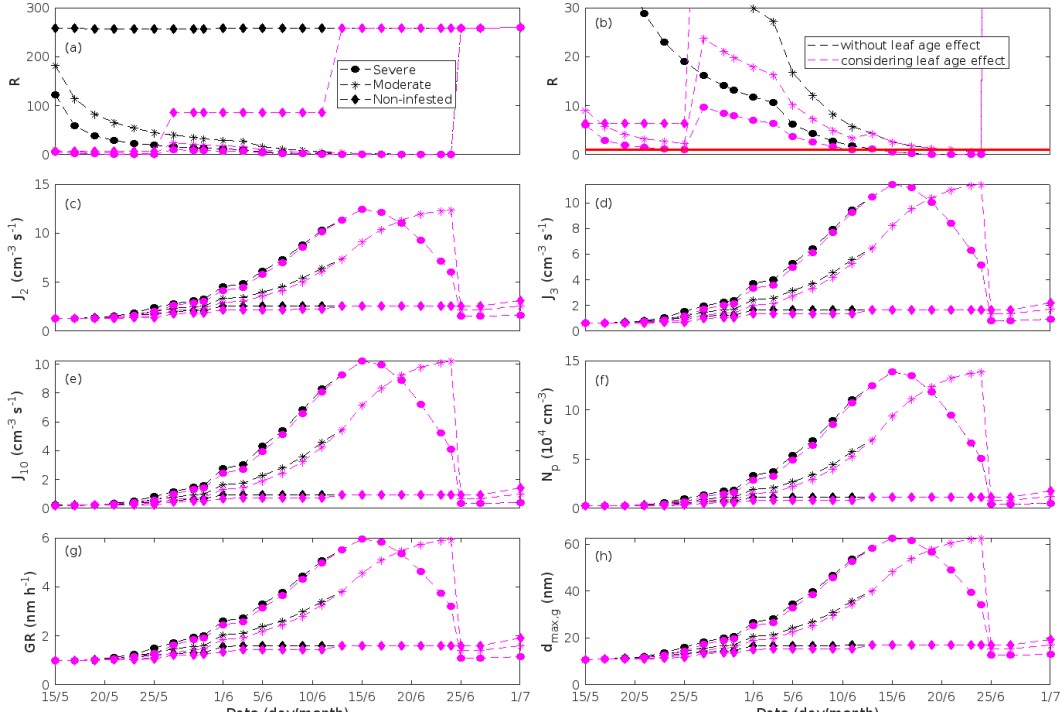

**Figure A2.** An oak stand infested with European gypsy moth larvae in comparison to a non-infested oak stand simulated with and without considering the impact of leaf age on the rates of emissions. **(a)** the ratios of isoprene-to-monoterpene carbon concentrations provided as a zoom in **(b)**, where the red line indicates R = 1. Formation rates of **(c)** 2, **(d)** 3 and **(e)** 10 nm particles. **(f)** number concentrations of formed particles, **(g)** growth rates of newly formed particles, and **(h)** the daily maxima diameter of the growing particle mode. Symbols mean the same as in Fig. A1.

**Simulations of pedunculate oak infected by oak powdery mildew and balsam poplar infected by rust fungi**

The emission rates used for simulations of oak and poplar, with and without pathogenic infection, were measured in the middle and beginning of September, respectively, in Estonia (Copolovici et al., 2014; Jiang et al. 2016). Representative photographs of control leaves indicate that the measured leaves were mature and without any visible signs of senescence (Copolovici et al., 2014; Jiang et al. 2016).

When leaves grow old, they eventually lose their ability to photosynthesise and produce isoprene (Monson et al., 1994; Schnitzler et al., 1997; Sun et al., 2012) and Guenther et al. (2012) e.g. assumed a reduction of 10 % in the emissions of isoprene from senescing leaves (compared to that of mature leaves). However, a reduction on such a scale (i.e. 10 %) is not sufficient to decrease the ratio of isoprene to monoterpene carbon concentration to less than one in our simulations of oak and poplar infected by fungi. The impact of leaf age was therefore tested during the period with stress by decreasing the emission rate of isoprene to such a degree that R was either just under 22.5 or just under 1. Since isoprene does not show an induced response in emission upon oak powdery mildew or rust infection, the emission rate of isoprene was reduced in simulations of both non-infected and stressed oak and poplar forest. The results are shown in Fig. A3-4.



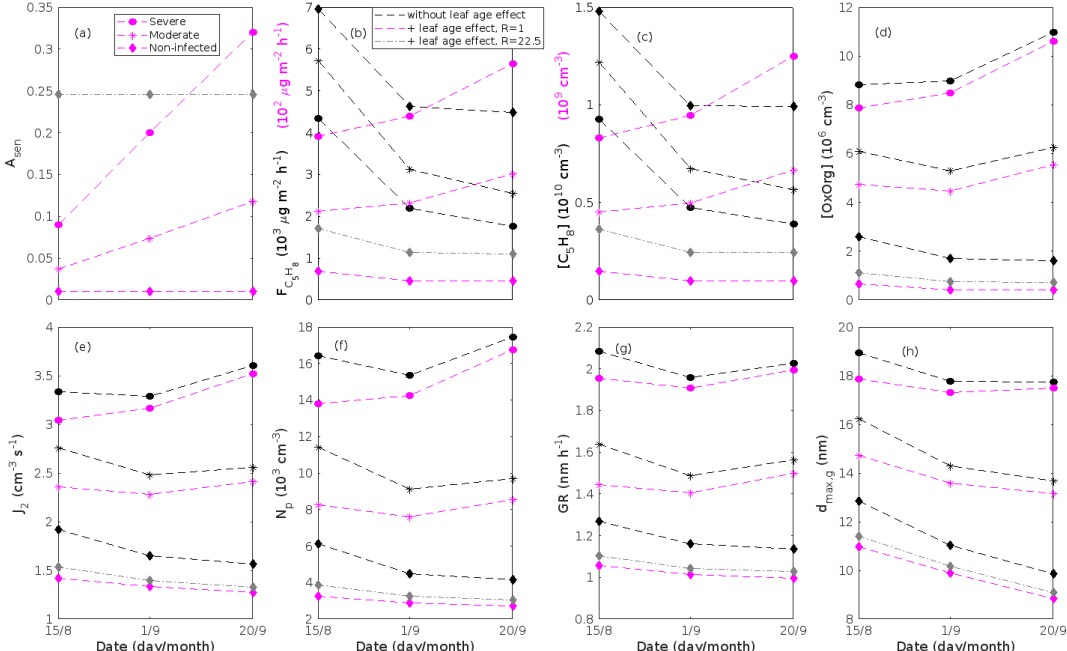

**Figure A3.** An oak stand infected by oak powdery mildew in comparison to a non-infected oak stand simulated with and without considering the impact of leaf age on the rates of isoprene emissions. **(a)** the fraction of isoprene emitted in comparison to simulations where the leaf age effect was not considered. For example, in order to reach R = 1 in simulations of a non-infected stand (magenta diamonds), the leaves are assumed to only emit 1% of isoprene compared to our default simulations of a non-infected stand (black diamonds in Fig. 7a). The syntax is equivalent to that of Guenther et al. (2012, p. 1476). **(b)** canopy emissions of isoprene and atmospheric concentrations of **(c)** isoprene and **(d)** OxOrg. The units provided in black in **(b-c)** are connected to black and grey data points, while the units in magenta in **(b-c)** are connected to magenta data points. **(e)** formation rate of 2 nm particles, **(f)** number concentrations of formed particles, **(g)** growth rates of newly formed particles, and **(h)** the daily maxima diameter of the growing particle mode. The values of other parameters during these simulations are the same as in Fig. 7. "Moderately" and "severely" refer to 30 % and 80 %, respectively, of the leaf area being infected by fungi by the 20th of September. Black makers are for simulations where the effect of leaf age was not considered (see Fig. 7 for what R is then), while magenta markers are for simulations where the emission of isoprene was reduced sufficiently for R to be just under 1. Grey diamonds are used to illustrate simulation results of a non-infected oak stand, where the emission of isoprene has been reduced sufficiently for R to be just under 22.5. Simulation results (independently of whether the effect of leaf age was considered or not) for "severe" are always indicated by circles, for "moderate" by asterisks, and for "non-infected" by diamonds.



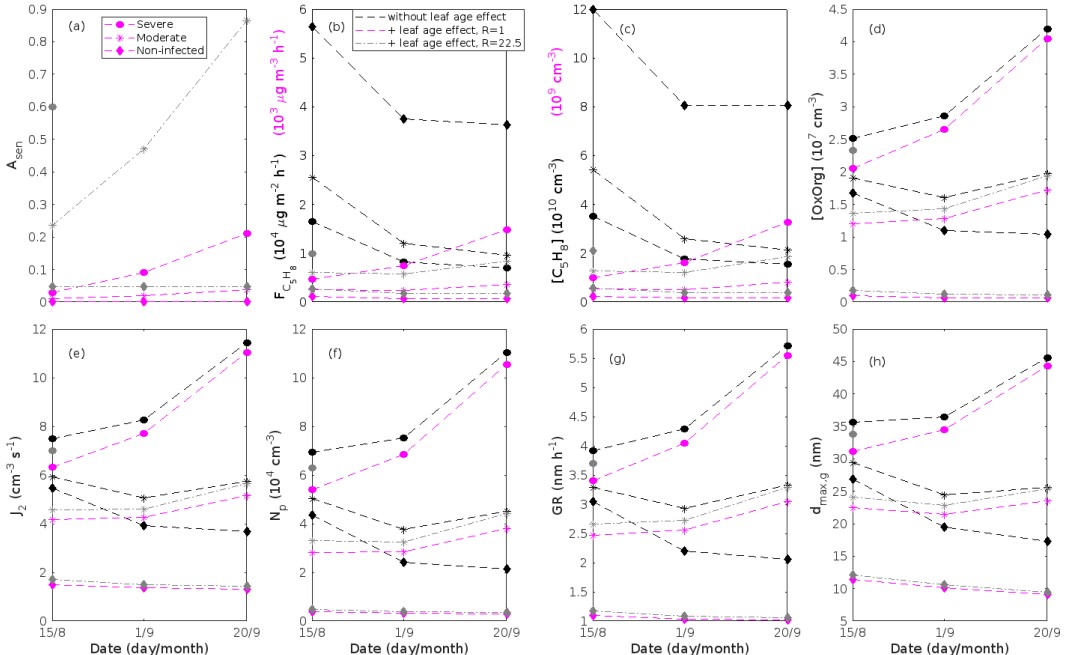

**Figure A4.** A poplar stand infected by rust in comparison to a non-infected poplar stand simulated with and without considering the impact of leaf age on the rates of isoprene emissions. **(a)** the fraction of isoprene emitted in comparison to simulations where the leaf age effect was not considered. **(b)** canopy emissions of isoprene and atmospheric concentrations of **(c)** isoprene and **(d)** OxOrg. The units provided in black in **(b-c)** are connected to black and grey data points, while the units in magenta in **(b-c)** are connected to magenta data points. **(e)** formation rate of 2 nm particles, **(f)** number concentrations of formed particles, **(g)** growth rates of newly formed particles, and **(h)** the daily maxima diameter of the growing particle mode. The values of other parameters during these simulations are the same as in Fig. 8. Grey markers are for simulations where the emission of isoprene was reduced sufficiently for R to be just under 22.5. Symbols are the same as in Fig. A3.

**Appendix B: Chemical reactions in the model**

We considered the following chemical reactions in our model:

OH + isoprene → product +γOxOrg$_{(g)}$, k=2.7·10$^{-11}$·e$^{(390/T)}$ cm$^3$·molecule$^{-1}$·s$^{-1}$, γ$_a$=0.03 % (B1)

O$_3$ + isoprene → product +γOxOrg$_{(g)}$, k=1.03·10$^{-14}$·e$^{(-1995/T)}$ cm$^3$·molecule$^{-1}$·s$^{-1}$, γ$_b$=0.01 % (B2)

OH + monoterpenes → product +γOxOrg$_{(g)}$, k=1.2·10$^{-11}$·e$^{(440/T)}$ cm$^3$·molecule$^{-1}$·s$^{-1}$, γ$_c$=1.7 % (B3)

O$_3$ + monoterpenes → product +γOxOrg$_{(g)}$, k=8.05·10$^{-16}$·e$^{(-640/T)}$ cm$^3$·molecule$^{-1}$·s$^{-1}$, γ$_d$=5.0 % (B4)

OH + methyl salicylate → product +γOxOrg$_{(g)}$, k=4.0·10$^{-12}$ cm$^3$·molecule$^{-1}$·s$^{-1}$, γ$_e$=10.0 % (B5)

OH + LOX → product, k=6.0·10$^{-11}$ cm$^3$·molecule$^{-1}$·s$^{-1}$ (B6)

OH + methanol → product, k=2.85·10$^{-12}$·e$^{(-345/T)}$ cm$^3$·molecule$^{-1}$·s$^{-1}$ (B7)

O$_3$ + DMNT, sesquiterpenes or oxidised sesquiterpenes → product + γOxOrg$_{(g)}$, k=1.2·10$^{-14}$ cm$^3$·molecule$^{-1}$·s$^{-1}$, γ$_f$=7.7 % (B8)

OxOrg$_{(g)}$ → OxOrg$_{(p)}$, k=CS$_{SA}$·0.5 (B9)

OH + SO$_2$ → H$_2$SO$_{4(g)}$, k=1.5·10$^{-12}$ cm$^3$·molecule$^{-1}$·s$^{-1}$ (B10)

H$_2$SO$_{4(g)}$ → H$_2$SO$_{4(p)}$, k=CS$_{SA}$ (B11)



T is temperature (K), p indicates "particle phase", $CS_{SA}$ is the condensation sink determined for sulfuric acid, while $\gamma_i$ are the
fractions of organic products which can partition to the particle phase (OxOrg). In practice, $\gamma_i$ are either reported HOM yields,
as defined in Ehn et al. (2014), or reported SOA yields / 2.2. SOA yields are decreased by a factor of 2.2 in order to account
for the fact that SOA yields represent mass yields, and not molar yields as it is the case of HOM yields. We utilised HOM
yields based on Jokinen et al. (2015) ($\gamma_a$, $\gamma_b$, $\gamma_c$, $\gamma_d$), Berndt et al. (2016) ($\gamma_c$), and Ehn et al. (2014) ($\gamma_d$). We used SOA yields
from Mentel et al. (2013) ($\gamma_e$, $\gamma_f$).
**Appendix C: Sensitivity tests**
**Table C1.** Constrained parameters for sensitivity tests and their range of values. Nine different sensitivity tests (ST1-9) were
conducted for all plant species and infections, where only one parameter was changed at a time. BHL is the planetary boundary
layer height, $PPFD_{max}$ is the daily maximum photosynthetic photon flux density, $T_{max}$ is the maximum daily temperature, LAI
is the leaf area index of non-infested leaves, $CS_{SA}$ is the condensation sink determined for sulfuric acid and $\gamma_{OxOrg}$ is the yield
of OxOrg. "HPB" and "SMEAR I" refer to simulations conducted in Hohenpeissenberg (i.e. oak and poplar) and SMEAR I
(i.e. birch) conditions, respectively.

| Sensitivity test no. | Parameter that changes | HPB | SMEAR I | Notes and references |
|---|---|---|---|---|
| ST1 | BLH (m) | 200 - 2500 | | Classical textbook example. |
| ST2 | $PPFD_{max}$ ($\mu mol\ m^{-2}\ s^{-1}$) | 200 – 2000 | 200 - 1600 | The lower limit is based on observations at the SMEAR I station, the upper on the theoretical clear sky maxima. |
| ST3 | $T_{max}$ (ºC) | 10 - 40 | 5 - 35 | Based on observations and the IPCC 2014 predictions of the regional temperature increase. |
| ST4 | LAI ($m^2\ m^{-2}$) | 1 - 9 | 0.5 – 4 | The upper limit is based on Tripathi et al. (2016). |
| ST5 | $[O_3]$ (ppb) | 35 - 100 | 20 - 70 | Naja et al. (2003), Ruuskanen et al. (2003). The upper end for HPB simulations is similar to the highest values which are observed in the Amazon where concentrations of isoprene can be very high (e.g. Pacifico et al., 2015) |
| ST6 | $[OH]_{max}$ (molec $cm^{-3}$) | $0.1\text{-}10\cdot10^6$ | $1\text{-}25\cdot10^5$ | Petäjä et al. (2009), Berresheim et al. (2000), Rohrer and Berresheim (2006). The lower limit has not been observed in HPB, but is included in order to test the impact of potential OH depletion on our results. |
| ST7 | $[H_2SO_4]_{max}$ (molec $cm^{-3}$) | $5\text{-}25\cdot10^6$ | $1\text{-}6.5\cdot10^6$ | Birmili et al. (2003), Kyrö et al. (2014). |
| ST8 | $CS_{SA}$ ($s^{-1}$) | $2\text{-}8\cdot10^{-3}$ | $0.2\text{-}3.5\cdot10^{-3}$ | Birmili et al. (2003), Vana et al. (2016), Kyrö et al. (2014). |
| ST9 | $\gamma_{OxOrg}$ | $0.1 - 2\cdot\gamma_{OxOrg,default}$ | | Ehn et al. (2014), Jokinen et al. (2015), Bianchi et al. (2019), McFiggans et al. (2019). |


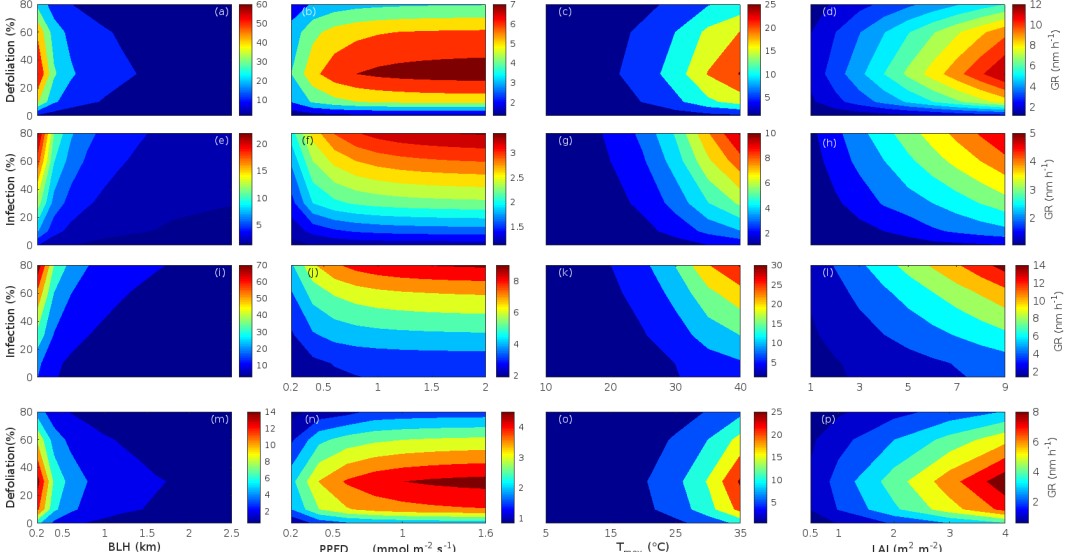

**Figure C1.** Impact of changed boundary conditions on the growth rate of small particles in non-infected and biotically stressed forest stands. The subplots correspond to those in Fig. 10, except the subplots here display growth rate, and not number of particles. Thus we refer to Fig. 10 for further explanations.

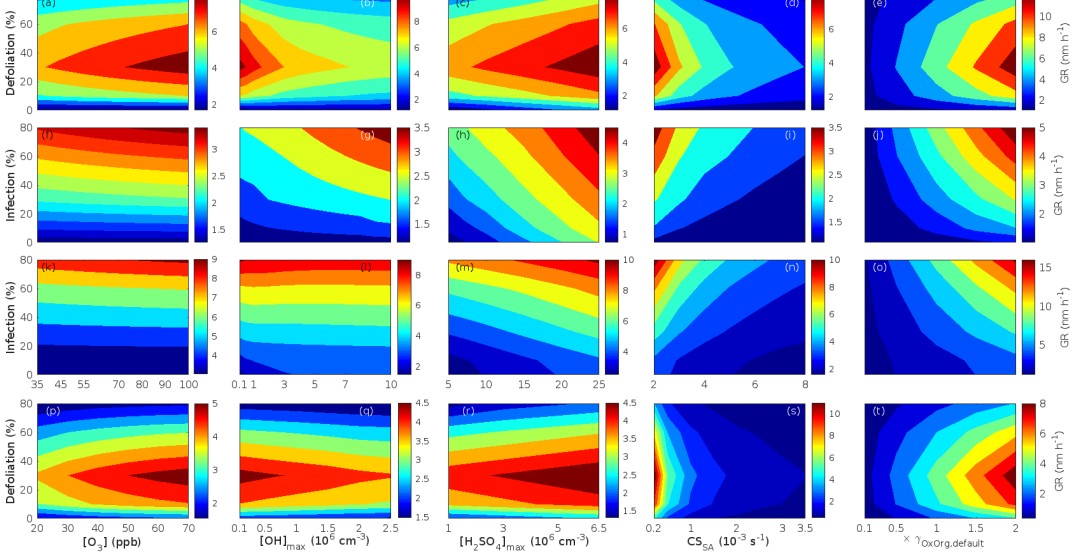

**Figure C2.** Impact of changed boundary conditions on the growth rate of small particles in non-infected and biotically stressed forest stands. The subplots correspond to those in Fig. 11, except the subplots here display growth rate, and not number of particles. Thus we refer to Fig. 11 for further explanations.