# Peer review of "Modelling the influence of biotic plant stress on atmospheric aerosol"

_Atmospheric Chemistry and Physics, 2021_

## Author Response (AR1)

This manuscript describes the development and application of a numerical model of biotic stress induced VOC emissions to examine their impact on atmospheric aerosol. The biotic stresses include pathogen infections (oak powdery mildew, poplar rust) and herbivore infestations (moth larvae). These stresses serve as useful examples of the various types of biotic stress and are an appropriate starting place for investigating biotic stress BVOC impacts since quantitative information is available for both simulating stress scenarios over a growing season and for estimating the VOC emission response. An atmospheric chemistry and aerosol dynamics model is used to examine the implications for particle production and growth. The authors find that the plant stresses can increase new particles by up to an order of magnitude and daily growth up to about 50 nm. They conclude that it can be more important to account for biotic plant stresses in models than variations in LAI, light and temperature. This effort is a valuable contribution to the existing literature on this topic and a good fit with ACP and should be considered for publication after the authors address the following issues.

>>We sincerely thank the reviewer for having taken the time to carefully read and review our manuscript and for providing constructive comments and suggestions, in a supportive manner, which will improve our work and manuscript.

It is well known that biotic stress can greatly modify BVOC emissions and that BVOC are a major source of new particles and SOA mass so the finding that stress "can" play an important role in controlling atmospheric aerosol is expected and agrees with previous studies that have speculated on this. What is important about this study is that it takes additional steps towards a quantitative representation of the processes controlling biotic stress in controlling biogenic SOA. However, it does include unsubstantiated conclusions including:

Line 777 and 814: "it can be more important to account for biotic plant stresses in models than significant variations in e.g. LAI, and temperature and light conditions, which are currently the main parameters controlling predictions of VOC emissions." This is misleading since it does not say important for what (i.e., is it important for any air quality or climate concerns?). The same argument could be made for wildfires, urbanization, hailstorms, and many other processes that they "can" be more important than LAI, light and temperature in determining BVOC emissions at a specific time and place but that does not mean that they ever are important for any air quality or climate simulations. The results of this study need to be put into context by considering the scale and frequency of these stresses when describing the impacts. The authors may not have the right model tool for quantifying this but they can at least discuss it and qualify their statements/conclusions. It should also be noted that factors such as light and temperature have been studied extensively on canopy to landscape scales, while the abiotic stresses have not, and enclosure studies often do not accurately represent what is observed on canopy to landscape scales. As a result, it is difficult to make a convincing case regarding the impact of biotic stress on SOA without some validation of these results on landscape/regional scales. The authors should discuss this current lack of validation and consider how it impacts their conclusions and perhaps they could outline what needs to be done to confirm these model results.

>>Thank you for this very good and important comment! About the statements at L777 and 814, we will add "...when predicting new particle formation and growth". Also, in the manuscript, we will clarify the conditions/scenarios where the quoted statement is believed to be true. You are correct that considering the scale and frequency of the studied stresses are important in order to properly evaluate the impact of them, thus we will include a discussion about this in Sec. 3.3. In Sec. 3.3, we will also mention facts such as those you mention in your comment (e.g., that we have limited evidence of the impact of stresses on canopy and landscape scales and that such are needed), emphasis that we are unable to validate our modelling results due to lack of observations, that the evidence for the importance of stress emissions based on various literature are mixed, and we will

also underline that robust representations of stress emissions are needed in order to not introduce errors into models.

[revised manuscript text omitted]
 models.". Due to overlap in the text, the following paragraph in the original Sec. 3.3 was delete "Vegetation is often subject to more than one type of stress simultaneously, which generally enhances the already induced emission response due to biotic plant stress (e.g. Blande et al., 2007; Vapaavuori et al., 2009; Holopainen and Gershenzon, 2010; Kivimäenpää et al., 2016; Ghimire et al., 2017). It has, for example, recently been shown that warming significantly amplifies the emission response due to autumnal moth feeding (Li et al., 2019). Trees which have been subject to herbivory also have a tendency to more often be severely attacked by fungi later during the summer, than trees which have not experienced defoliation (e.g. Marçais and Desprez-Loustau, 2014).". In Sec. 3.3 we deleted "Together with the facts that insect outbreaks and fungal diseases generally are expected to increase in both frequency and severity in the future (Cannon, 1998; Bale et al., 2002; Harrington et al., 2007; Pautasso et al., 2012; Boyd et al., 2013), our findings underline the urgency of accounting for biotic plant stress emissions in numerical models.".

Line 784 and 817: " findings underline the urgency of accounting for biotic plant stress emissions in numerical models". These findings will not make this urgent unless the authors can provide some indication that this happens on a scale or frequency that is important. Include more

discussion/recognition regarding what is needed to show the significance for actual scenarios and if it is unknown then perhaps suggest what field measurements are needed to get to this.

>>You are correct, and we will add this to the discussion in Sec. 3.3 (see reply above), and also modify the statement (L784+817) accordingly.

CHANGES TO MANUSCRIPT: See changes made in the point above.

The model approach used for this study may be reasonable for the objectives of the study but the declaration (Line 287) that "we constrained the concentrations of atmospheric oxidants within the model, since it is unreasonable to assume that they can be accurately predicted" is not justified. If it is unreasonable to assume oxidants can be accurately predicted, then it is equally unreasonable to assume that SOA can be accurately predicted or even that stress BVOC emissions can be accurately predicted. This is not to say that the authors need to calculate oxidants in their model, it is just that is not the reason that should be given for not calculating oxidants. Instead, the authors just need to demonstrate that calculating oxidants is not necessary for the objectives of this study. Another issue with the modeling the description of how it is implemented. I assume that this is a 0D model but I don't see where that is described.

>>Fair point, thank you! We will reformulate the sentence and also refer to Fig. 11 in which it has actually been demonstrated that calculating oxidants is not necessary for the objectives of the study. Indeed the model is a 0D box model and we will specify this in Sec. 2.

CHANGES TO MANUSCRIPT: "Similarly to previous atmospheric modelling studies of herbivory (Bergström et al., 2014; Douma et al., 2019), we constrained the concentrations of atmospheric oxidants within the model, since it is unreasonable to assume that they can be accurately predicted. Many studies do, for example, report a large (up to at least 89 %) missing sink of OH, especially in forested areas (e.g. Di Carlo et al., 2004; Sinha et al., 2010; Mogensen et al., 2011, 2015; Nölscher et al., 2012, 2016; Zannoni et al., 2016; Praplan et al., 2019) and studies above the Amazonian rainforest, furthermore, indicate that isoprene can recycle OH with a varying efficiency of 40-120 % (Lelieveld et al., 2008; Taraborrelli et al., 2012). In our simulations, the default daily maximum concentration of OH is therefore fixed to $6 \cdot 10^6$ molec cm-3 (Petäjä et al., 2009) and $8 \cdot 10^5$ molec cm-3 (calculated using observed summertime UVB radiation from the SMEAR I station and the proxy presented by Petäjä et al. (2009)) for simulations of Hohenpeißenberg and Lapland, respectively (Table 2). The daily pattern of the OH concentration then follows the solar zenith angle. The concentration of ozone is kept constant to a value of 45 ppb (Naja et al., 2003) and 30 ppb (Ruuskanen et al., 2003) for simulations of oak and poplar (Hohenpeißenberg conditions) and mountain birch (SMEAR I conditions), respectively (Table 2). In reality, the atmospheric oxidant concentration can, however, decrease or increase depending on changes in the concentrations of individual specific VOCs (Table 3). The impact of changing oxidation concentrations on our simulation results was therefore also tested (Sec. 3.2, Fig. 11a-b,f-g,k-l,p-q)." was reformulated to "Similarly to previous atmospheric modelling studies of herbivory (Bergström et al., 2014; Douma et al., 2019), we constrained the concentrations of atmospheric oxidants within the model, though in reality, the concentration of atmospheric oxidants can decrease or increase depending on changes in the concentrations of individual specific VOCs (Table 3). This was done, partly because it is difficult to accurately predict the concentration of oxidants (e.g. Di Carlo et al., 2004; Sinha et al., 2010; Mogensen et al., 2011, 2015; Nölscher et al., 2012, 2016; Zannoni et al., 2016; Praplan et al., 2019; Lelieveld et al., 2008; Taraborrelli et al., 2012), and partly because accounting for varying oxidant concentrations is not necessary for the objectives of our study (Sec. 3.2, Fig. 11a-b,f-g,k-l,p-q). Thus, in our simulations, the default daily maximum concentration of OH is therefore fixed to $6 \cdot 10^6$ molec cm-3 (Petäjä et al., 2009) and $8 \cdot 10^5$ molec cm-3 (calculated using observed summertime UVB radiation from the SMEAR I station and the proxy presented by Petäjä et al. (2009)) for simulations of Hohenpeißenberg and Lapland, respectively (Table 2). The daily pattern

of the OH concentration then follows the solar zenith angle. The concentration of ozone is kept constant to a value of 45 ppb (Naja et al., 2003) and 30 ppb (Ruuskanen et al., 2003) for simulations of oak and poplar (Hohenpeißenberg conditions) and mountain birch (SMEAR I conditions), respectively (Table 2).". In Sec. 2 Materials and methods, we reformulated "We constructed a model that includes modules for emissions of VOCs from stress-free and biotically stressed tree species (Sec. 2.4), ..." to "We constructed a 0D box model that includes modules for emissions of VOCs from stress-free and biotically stressed tree species (Sec. 2.4), ...".

**Anonymous Referee #2**

"This manuscript provides an interesting new model study on the possible impact of stress induced BVOC on aerosol formation. They provide interesting case studies to illustrate the possible impacts of different stresses, including providing suggestions for the timing of infection dynamics, and impacts of stresses on LAI as well as emissions."

>>We thank the reviewer for taking the time to carefully read our submission and for providing insightful comments for improvement of the manuscript.

"In reading the manuscript I find that my main concerns are very similar to those raised by Ref #1: that the authors overplay the need and possibility of including such SIE emissions into atmospheric models. Indeed, it was amusing to read on L288 that it is "unreasonable to assume that [oxidants] can be accurately predicted", but still the authors advocate adding another layer of hugely uncertain SIE algorithms to Earth System models."

>>We refer to our replies to Ref#1's similar concerns.

CHANGES TO MANUSCRIPT: See above.

"This paper (and its predecessors) does make a strong case that SEI can be important in certain situations, but we are far from knowing how important that are in the real atmosphere. The great need is for new observations to constrain the role of such SEI in SOA formation, and for new ideas on how such emissions could be realistically included in 3-D models. What would it take for example to try to simulate gypsy-moth frequency on global or even European scales? Do we have any chance to tackling such issues in a robust way in the next 5-10 years?'

On the same theme, the authors seem to play down the possibility that SIE emissions are not that important. L467 does cite the work of Ylivinkka et al 2020 who did not find any significant sign of SIE being important for aerosols, but another important but uncited study is that of Berg et al (2013), which as well as providing one of the first 3-D modeling studies of SIE emissions, also suggests that the impacts of beetle-attacks on SOA are quite small in comparison to the impacts of wildfires. As far as I can see, the evidence is mixed, but this reinforces the need for observational studies to sort out the issues."

>>About the first part: we completely agree with the reviewer – accounting for stress-induced emissions, and especially emissions induced in response to biotic plant stress, and multiple co-occurring stresses, in large scale models, in a robust manner, is no small thing, and not something which will happen overnight. It is something which in practice will happen in steps. Our manuscript is one such step, since it goes further with the *quantitative* representation. For example, we quantitatively account for the impact of the degree of stress on emission rates of VOCs, emission spectrum of VOCs, and formation and growth processes of aerosols. Also, as the first ones, we take the dynamics of herbivory and fungi into account.

About the second part: thanks for giving the heads-up about the Berg paper! We will include a reference to it in the intro (around L85-92 when results from Bergström et al. (2014) and Joutsensaari et al. (2015) are described) and when doing so, emphasis that there is a possibility that biotic plant stress emissions are not necessary for describing NPF in the atmosphere. We will state that indeed the evidence is mixed.

CHANGES TO MANUSCRIPT: To the introduction, we added: "Unfortunately, such quantitative estimates are currently very scarce, connected with a large degree of uncertainty, and not necessarily reaching the same conclusions. For example, Berg et al. (2013) used bark beetle-induced monoterpene emissions responses and several years of bark beetle-induced tree mortality data from western North America as input to a global model in order to investigate the impact of bark beetle attacks on regional secondary organic aerosol (SOA) formation. The authors found that the concentration of SOA might increase regionally by up to 40 % or 300 % in case of bark beetle attacks on lodgepole pine and spruce trees, respectively (Berg et al., 2013). At the same time, Berg et al. (2013) concluded that the enhancement in the concentrations of SOA are in most cases small in comparison to the impacts of wildfires on total organic aerosol in western North America." and "Thus, the degree of necessity of considering biotic plant stress emissions for predictions of new particle formation in the atmosphere is still uncertain, and therefore there is a great need for field observations of different scales to constrain and quantify the role of stress emissions in SOA formation, but also for innovative model approaches which improve the quantitative representation of the emissions.".

"I must admit I also found the results section (p14-27) rather long and detailed considering the large uncertainties and the simplicity of the (unexplained) modeling scheme used. Lots of details are given about growth rates, and comparison with literature values, but the assumption of prescribed oxidants makes these predictions very difficult to interpret. I think the authors need to bear in mind more strongly that they are presenting a very conceptual model, and not a real atmospheric simulation. I think their modeling is sufficient to make the case that SEI may be important, and deserve much closer attention, but I would try to be more concise, possibly by tabulate more of the results."

>>We had tried to be clear in our original manuscript that our modelling study is of a conceptual character and the modelling results should therefore also be treated accordingly. We will clarify this further/state this stronger in the manuscript. In our opinion it is useful for the reader that the modelling results are compared in detail to observations (like it is done in the result sections) in order to put our findings into perspective, as long as the model description is clear, and the uncertainties and limitations of the study are discussed, which we have already tried to do in the manuscript, and which we will further improve in response to the reviewers' comments.

CHANGES TO MANUSCRIPT: In Sec. 3.1, we added: "We underline that our modelling study is of a conceptual character and that the modelling results should therefore also be treated accordingly. Modelling results are compared in detail to observations in the sections below in order to put our findings into perspective."

"Despite these misgiving, this paper is suitable for ACP since it makes a new contribution to the existing literature on this interesting and potentially very important topic. It should be considered for publication after addressing the various issues raised by the referees.

Detailed comments:

L51, and elsewhere. I think it would read better if the authors used the terms "increased" rather than "induced" in many places. Here I would say that "emissions of other (induced) VOC are greatly

increased" (I am not sure "greatly induced" is correct English anyway.). Actually, a small explanation of constitutive and induced VOC might help readers for which such terms are not obvious."

>>About induced vs increased: you are right that at certain places (like L51), it would make more sense to use "increased" instead of "induced". At certain other places, where the emissions of specific VOCs are only emitted in response to plant stress, it would make better sense to stick with "induced". Thus, we will go through the manuscript and correct the wording when appropriate. About constitutive and induced VOCs: OK, this is a good point. On L46 we will add: "Many plants emit VOCs constitutively, i.e. that they emit VOCs regardless of the experience of stress. Biotic plant stress (i.e. stress caused to a plant by living species such as e.g. herbivores and pathogens) is known to substantially alter both the rates of emission and spectrum of VOCs emitted constitutively (Holopainen and Gershenzon, 2010; Niinemets, 48 2010; Niinemets et al., 2013; Faiola and Taipale, 2020). Emissions of VOCs which are increased, or started to be emitted, in response to plant stress are referred to as induced plant volatile emissions."

CHANGES TO MANUSCRIPT: "induced" was changed to "increased" at the appropriate places. "Many plants emit VOCs constitutively, i.e. that they emit VOCs regardless of the experience of stress." was added on L46, while "Emissions of VOCs which are increased, or started to be emitted, in response to plant stress, are referred to as induced plant volatile emissions." was added on L52-53, and "Biotic plant stress (i.e. stress caused to a plant by living species such as e.g. herbivores and pathogens) is known to substantially alter both the rates of emission and spectrum of emitted VOCs (Holopainen and Gershenzon, 2010; Niinemets, 2010; Niinemets et al., 2013; Faiola and Taipale, 2020)." was re-written to "Biotic plant stress (i.e. stress caused to a plant by living species such as e.g. herbivores and pathogens) is known to substantially alter both the rates of emission and spectrum of VOCs emitted constitutively (Holopainen and Gershenzon, 2010; Niinemets, 2010; Niinemets et al., 2013; Faiola and Taipale, 2020)." on L46-49.

"L61. The phrase "until now" suggests that the current paper is providing the "consistent mechanism" mentioned in this sentence. I don't think the authors mean that. Re-phrase."

>>You are correct that this was not the intention with the wording. We reformulate the sentence to "Thus, no consistent mechanism for the emissions of VOCs from plants under stress exist.".

CHANGES TO MANUSCRIPT: The sentence was reformulated accordingly.

"L80-81. Add "over short periods at least""

>>OK, very good, we will add that to the sentence.

CHANGES TO MANUSCRIPT: "over short periods at least" was added to the sentence.

"L109- Add references for the statements made about these insects"

>>OK, we will add the following references:

Klemola, T. et al., Oecologia, 141, 47-56, https://doi.org/10.1007/s00442-004-1642-z, 2004

Ammunét, T. et al., Ecography, 34, 848-855, doi: 10.1111/j.1600-0587.2011.06685.x, 2011.

https://www.cabi.org/isc/datasheet/31807#tohostsOrSpeciesAffected

CHANGES TO MANUSCRIPT: The references were added to the section.

"L114 - define vast areas. More generally, what percentage of national forest cover do the authors think are affected by these stressors?"

>>Good point. We add "Several thousands of square kilometres of birch forests have previously been reported to become defoliated due to just a single outbreak of autumnal moth in Fennoscandinavia (Tenow 1975; Nikula 1993), while gypsy moth, in North America alone, is estimated to have defoliated >95 million acres of forest during years 1920 to 2020 (Coleman et al., 2020).".

With the references being:
Tenow, O., Zoon, 3, 85-110, 1975.
Nikula, A., Animals as Forest Pests in Finnish Lapland, vols. 22-29, 1993.
Coleman et al., Forest Insect & Disease Leaflet 162 April 2020, US Forest Service.

CHANGES TO MANUSCRIPT: The suggested sentence was added.

"L134. What is "instar"?"

>>An instar is a developmental larval stage. On L112, we reformulate the sentence to "Both sexes have five larval stages (instar), though female gypsy moths have six."

CHANGES TO MANUSCRIPT: The sentence was reformulated in accordance with our reply.

"L142. The caption is confusing with e.g. "(c). d-g, then on explanation of (d) on L143. I suggest omitting "a-c" and "d-g" bits."

>>OK, we will follow your suggestion.

CHANGES TO MANUSCRIPT: "a-c" and "d-g" where omitted.

"L186. Give Institute for pers.comm."

>>OK, we add "University of Helsinki".

CHANGES TO MANUSCRIPT: We added "University of Helsinki" to the sentence.

"L212. Why use 25C as the base, instead of 30C as used in most BVOC papers?"

>>As such, there is no specific reason for this, except that three out of the four papers which the emission responses are based on, measured the emission rates at 25C. To this should be added that 25C is also very often used as the standard condition in the literature. The important thing is that either by standardising the emission to 25 or 30C will not change the result.

CHANGES TO MANUSCRIPT: Following our reply, nothing was changed in the manuscript related to this point.

"L212. Do you mean 1-sided or projected LAI, or something else?"

>>One-sided LAI. We will add the information to the sentence.

CHANGES TO MANUSCRIPT: We added "one-sided" to the sentence.

"L247. The symbol D is usually used in the BVOC literature for foliar biomass. Could another symbol be found for degree of stress?"

>>We change it to Æ.

CHANGES TO MANUSCRIPT: D was changed to Æ throughout the manuscript.

"L247 On: I found Table confusing in many respects. On L249 it is stated that for some species a factor 0.57 has been used, but on L250 others have been "upscaled", whatever that means. Emissions rates include per m2 terms, but are these m2 LAI or m2 ground-area? Clarify."

>>OK. About the factor of 0.57: On L219-225 we inform that emissions from oak and poplar are multiplied by a factor of 0.57, because this is commonly what is done when the used emission rates are retrieved from leaf level measurements and your model does not include a full canopy environment scheme. Since the factor of 0.57 is included to compensate for the lack of canopy environment, we did not want to include it within the reported emission factors. The sentence (L249) "The emission factors, listed for oak and poplar in the table, have not been downscaled (by a factor of 0.57)..." was therefore included to clarify this fact. We will reformulate the sentence to "The emission factors for oak and poplar are presented without the downscaling by a factor of 0.57 (see bulk text in Sec. 2.4).". About the upscaling: three of the four papers we used to retrieve the emission factors measured the emissions from mature trees. Only Yli-Pirilä et al. (2016) measured on seedlings. Since the leaves of mountain birch seedlings are smaller and lighter than leaves growing on mature mountain birches, an upscaling of the emission rates and factors needed to be done. This is specified on L226-228, and then mentioned again in the table caption on L250- ("...but the emission factors for mountain birch, listed here, have been upscaled in order to represent the emissions from mature trees. Thus, LMAf is the fraction of the leaf mass area of leaves growing on mature mountain birch / growing on mountain birch seedlings") in order to clarify this. We will delete "but the emission factors for mountain birch, listed here, have been upscaled in order to represent the emissions from mature trees" and rewrite "Thus, LMAf is the fraction of the leaf mass area of leaves growing on mature mountain birch / growing on mountain birch seedlings" to "LMAf is the fraction of the leaf mass area of leaves growing on mature mountain birch / growing on mountain birch seedlings, which is included so that the emission factors for mountain birches are representative for mature trees.". The emission factors in the table are provided in unit per m-2 one-sided LAI and this will be specified in the table caption.

CHANGES TO MANUSCRIPT: The sentence "The emission factors, listed for oak and poplar in the table, have not been downscaled (by a factor of 0.57)..." was reformulated to "The emission factors for oak and poplar are presented without the downscaling by a factor of 0.57 (see bulk text in Sec. 2.4).". The sentence "but the emission factors for mountain birch, listed here, have been upscaled in order to represent the emissions from mature trees" was deleted. The sentence "Thus, LMAf is the fraction of the leaf mass area of leaves growing on mature mountain birch / growing on mountain birch seedlings" was replaced with "LMAf is the fraction of the leaf mass area of leaves growing on mature mountain birch / growing on mountain birch seedlings, which is included so that the emission factors for mountain birches are representative for mature trees.". It was furthermore specified in the caption of Table 1 that the emission factors, in the table, are provided in unit per m-2 one-sided LAI.

"Table 1 in general is very hard to interpret, since so many equations and factors are used. It would be very helpful to add another column or two with some kind of standard or typical emission rate, so that one sees emissions factors in ug/m2/s at say 25C, LAI 2.5, 1000 umol/m2/s PPFD, and some degree of stress."

>>This is a good idea. Since the table is already very comprehensive, and in order to not confuse the people which are reading the manuscript fast (so that they do not think that the simulations were only carried out at one specific degree of stress, because this was of course not the case), it is perhabs more suitable to add an additional table in the Appendix with such info. Then there would also be the possibility to write out the emission factors at a few different degrees of stress instead of just one. Thus, we will add an additional table to the appendix where the emission factors at different degrees of stress are written out.

CHANGES TO MANUSCRIPT: A table (Table A1) was included in an additional appendix (Appendix A). The table includes the emission factors at stress-free conditions and at four different degrees of stress. Since we now included an additional appendix, we have also changed the appendix numbers accordingly (e.g. the previous Appendix A is now Appendix B, and the previous Table A1, is now Table B1, and so on).

"Figure 4. Why different styles (line versus scatter) for (a) and (b)?"

>>Lines were used for (b), because we used monthly averaged max and min temperatures for central Europe simulations. Point markers were used in (a), because we used actual daily observations from SMEAR I.

CHANGES TO MANUSCRIPT: Nothing was changed in the manuscript related to this point, since the information that monthly averaged min and max temperatures for Germany and daily observations from SMEAR I are used were already mentioned in Sec. 2.5.

"L286 on, Sect 2.6. As noted by Ref #1, there is no information given on the type of model used. As BLH is used as a parameter, we can guess that it is a box model, but it is remarkable to omit both a description of the model, and any information on whether the model used has any abilities to reproduce SOA formation at all."

>>We refer to our reply to Ref#1's similar concern. About the model's ability to reproduce SOA formation: the model has been thoroughly tested by constraining and validating it with observations from the SMEAR II station in Hyytiälä, Finland. The analysis is, however, not included in the manuscript, because the manuscript is already very long and compact. However, another manuscript which does include this analysis is currently in preparation.

CHANGES TO MANUSCRIPT: In Sec. 2 Materials and methods, we reformulated "We constructed a model that includes modules for emissions of VOCs from stress-free and biotically stressed tree species (Sec. 2.4), ..." to "We constructed a 0D box model that includes modules for emissions of VOCs from stress-free and biotically stressed tree species (Sec. 2.4), ...".

"L329 and generally. The only oxidants considered are ozone and OH. The NO3 radical is known to be an important oxidant for SOA formation; why is this not considered?"

>>The referee is correct that NO3 is generally speaking important for SOA formation. However, NO3 was not included, since we only simulated the emission and atmospheric processes during day time, and NO3 is not relevant during daytime, due to it's very low daytime concentrations.

CHANGES TO MANUSCRIPT: We added the following sentence to Sec. 2.6: "NO3 was not considered, since emission and atmospheric processes were only simulated during day time, when the concentration of NO3 is insignificant."

OTHER CHANGES: When revising the manuscript, we noticed that Eq. 5 should not have been included and thus it was deteled. After Eq. 4 we added "where CS (s-1) is the condensation sink. When used together with Eq. 3, the value of CS is that of sulfuric acid.". This error has little influence on the actual simulation results and none on the conclusion.

---

## Author Response (AR2)

Major comment:

1. Sect 2, L117 onwards (line numbers refer to annotated ATC manuscript).

The "revised" model description consists of little more than saying "We constructed a 0D box model". The text still gives no evidence that the model has any ability to predict SOA in realistic conditions at all. This omission is justified in the "answers-to-referee" document with the comments that the document is already very long, and that much evaluation has indeed been done, but the publication is only at the in-preparation stage. Even though this 2nd paper is still in the preparation stage, I think the authors need to mention that some evaluation work has been done. (Actually, it is regrettable that SOA results from a model are presented before the model evaluation is presented.)

The comment that this explanation was omitted because the manuscript was already very long is odd given that the authors use ca. 14 pages on presenting results from this non-verified model. As I noted in my original review, I thought the results section was over-long, and that could have condensed given the simplicity of the model.

I would encourage the authors to add some summary of model performance, either in the main text or in the Supplementary.

As a detail concerning model description, the term BLH is used, but not its usage.

Thank you for this important comment. It would seem that our manuscript in its current stage has caused a misunderstanding. Thus, it is important to clarify that our work did not deal with SOA formation, but rather the influence of organic compounds on nucleation mode aerosol dynamics - more specifically on the evolution of aerosol number size distribution via NPF and subsequent particle growth. In order to prevent such a misunderstanding, a clarification has now been added to Sec. 2.

The first author has gone carefully through the model and found a few pieces of information about the model which were not already mentioned in the ~11 page long model description. Thus, additional model description has been added to Sec. 2, 2.4 and 2.6.

Regarding model evaluation, we have considered the point of the reviewer by now mentioning that evaluation of the model's ability to replicate organic condensation has been done for stress-free cases, and that the descriptions of several of the individual processes have been evaluated separately in earlier studies by either us or by other researchers (added to Sec. 2). Since SOA is primarily in the accumulation mode - which was not simulated in this work - it is clear that we cannot show any validation of the model related to predictions of SOA. SOA is additionally typically measured by AMS, and the AMS has a lower cutoff size of ~50 nm, thus there does not exist any SOA data for the sizes we have modelled.

Changes to the manuscript:

- In Sec. 2 we add: "The model is programmed in Fortran and the time step for each module is 60 s. The model was simulating one day at a time. The purpose of our study was to simulate the influence of low-volatility organic compounds on the evolution of aerosol number size distribution via new particle formation and subsequent particle growth. Thus, we were interested in changes in the nucleation mode dynamics, and not in aerosol mass concentration. The individual processes included in the model (Sec. 2.1-2.7) are aimed to imitate our best mechanistic understanding of those processes. The descriptions of several of the individual processes have been evaluated separately in earlier studies by either us or by other researchers (following the references provided

in the sections below). The model's ability to reproduce the influence of organic compounds on aerosol formation and growth in biotically stress-free conditions has furthermore been tested by constraining and validating the model with observations from the SMEAR II station (the Station for Measuring Ecosystem-Atmosphere Relations II) in Hyytiälä, Finland (Taipale, in prep)."

- In Sec. 2.4 we add "It was assumed that the emitted VOCs were instantaneously evenly distributed within the mixing volume, which is defined by the boundary layer height.".

- In Sec. 2.6 we add: "The chemistry was solved by the ordinary differential equation solver DLSODE (Radhakrishnan and Hindmarsh, 1993), and it was assumed that the concentrations of all considered atmospheric molecules were homogeneously distributed within the mixing volume.".

Minor comments:

2. Fig. 1. The caption is still a little confusing. One usually puts the list label (here a-h) before the explanation, thus (a) first text, (b) second text etc., but here the authors puts (a)-(g) after the texts, and "h" (inconsistently without parentheses) before the last text.
OK, we changed it according to the reviewer's suggestion.

3. Table 2: a) BHL in caption should be BLH, b) it would be good to add temperatures to this table also, so that all key parameters are included.
Thanks for giving the heads-up about the typo! In principle, it is a good idea to have all key parameters included in the table (and that's also what we have aimed at), but temperature is not included, because the daily maximum and minimum temperatures are not constant throughout the simulated period – which is opposite to all the parameters listed in Table 2. That's also why Fig. 4 is presented immediately after Table 2.

4. Sect 2.6: L307 The sentence starting "Similarly to previous atmospheric modelling studies of herbivory (Bergström et al., 2014; Douma et al., 2019), we constrained the concentrations..." is misleading I think. The Bergstrom et al paper uses explicit calculations of OH, O3 and NO3 from the gas-phase photochemical scheme of a 3-D CTM, and the Douma et al paper uses concentrations from the MLC-CHEM 1-D model. In both cases the SIE compounds themselves do not affect the calculated oxidant concentrations, but the type of "constraints" used in these papers are driven by chemical mechanisms, sensitive to meteorology and BVOC emissions, and thus of a different type to the very simplified system used by Taipale et al.
OK. We reformulate the sentence "Similarly to previous atmospheric modelling studies of herbivory (Bergström et al., 2014; Douma et al., 2019), we constrained the concentrations of atmospheric oxidants within the model, though in reality, the concentration of atmospheric oxidants can decrease or increase depending on changes in the concentrations of individual specific VOCs (Table 3)." to "We constrained the concentrations of atmospheric oxidants within the model, though in reality, the concentration of atmospheric oxidants can decrease or increase depending on changes in the concentrations of individual specific VOCs (Table 3)."

---

## Author Response (AR3)

Dear Editor,

Yes, you are correct that "prescribed" would be a better word and we have now changed it accordingly.